# Stomatin modulates adipogenesis through the ERK pathway and regulates fatty acid uptake and lipid droplet growth

Shao-Chin Wu [1], Yuan-Ming Lo[2], Jui-Hao Lee[3], Chin-Yau Chen[4], Tung-Wei Chen[1], Hong-Wen Liu[5], Wei-Nan Lian[2], Kate Hua[6], Chen-Chung Liao[6,7], Wei-Ju Lin[6], Chih-Yung Yang[8], Chien-Yi Tung [6✉] & Chi-Hung Lin [1,2,6,9✉]

Regulation of fatty acid uptake, lipid production and storage, and metabolism of lipid droplets (LDs), is closely related to lipid homeostasis, adipocyte hypertrophy and obesity. We report here that stomatin, a major constituent of lipid raft, participates in adipogenesis and adipocyte maturation by modulating related signaling pathways. In adipocyte-like cells, increased stomatin promotes LD growth or enlargements by facilitating LD-LD fusion. It also promotes fatty acid uptake from extracellular environment by recruiting effector molecules, such as FAT/CD36 translocase, to lipid rafts to promote internalization of fatty acids. Stomatin transgenic mice fed with high-fat diet exhibit obesity, insulin resistance and hepatic impairments; however, such phenotypes are not seen in transgenic animals fed with regular diet. Inhibitions of stomatin by gene knockdown or OB-1 inhibit adipogenic differentiation and LD growth through downregulation of PPAR$_\gamma$ pathway. Effects of stomatin on PPAR$_\gamma$ involves ERK signaling; however, an alternate pathway may also exist.

[1] Institute of Biophotonics, National Yang Ming Chiao Tung University, Taipei, Taiwan. [2] Institute of Microbiology and Immunology, National Yang Ming Chiao Tung University, Taipei, Taiwan. [3] Taiwan International Graduate Program in Molecular Medicine, National Yang Ming Chiao Tung University and Academia Sinica, Taipei, Taiwan. [4] Department of Surgery, National Yang Ming Chiao Tung University Hospital, I-Lan, Taiwan. [5] Chong Hin Loon Memorial Cancer and Biotherapy Research Center, National Yang Ming Chiao Tung University, Taipei, Taiwan. [6] Genomics Center for Clinical and Biotechnological Applications, Cancer Progression Research Center, National Yang Ming Chiao Tung University, Taipei, Taiwan. [7] Metabolomics-Proteomics Research Center, National Yang Ming Chiao Tung University, Taipei, Taiwan. [8] Department of Education and Research, Taipei City Hospital, Taipei, Taiwan. [9] Department of Biological Science and Technology, National Yang Ming Chiao Tung University, Hsinchu, Taiwan. ✉email: Cytung@nycu.edu.tw; linch@nycu.edu.tw

Stomatin is an ancient, widely expressed, oligomeric, monotopic integral membrane protein that is associated with cholesterol-rich membrane microdomains/lipid rafts[1,2]. The gene was first identified as the causal factor of Overhydrated Hereditary Stomatocytosis (OHSt) disease, and was therefore named "stomatin"[3,4]. However, stomatin knockout mouse was viable and did not show stomatocytosis phenotype[5]. So, besides being a major component of the lipid rafts, the functions of stomatin are largely unclear.

We have previously reported that stomatin, with its unique molecular topology, promoted cell-cell fusion by forming molecular assembly that recruited fusogenic protein to the appositional plasma membranes[6] as a prelude to cell-cell fusion. In addition to regulations of fusion events, stomatin can also interact with various plasma membrane proteins residing within the lipid rafts and modulate their activities. For examples, stomatin can regulate the transport activities of anion exchanger 1 (AE1)[7], glucose transporter (GLUT1)[8] and water channel aquaporin-1 (AQP1)[9], or activity of acid-sensing ion channel (ASIC) family[10,11]. Whether such regulations are mediated by stomatin's scaffolding effects that change the biophysical properties of the lipid rafts or by direct protein-protein interactions between stomatin and the effector molecules, are currently unknown.

It is also unclear if and how stomatin is involved in regulating signal transduction which is initiated by complex molecular interactions between ligands, receptors and signaling molecules that may occur in the vicinity of lipid rafts. It has been speculated that the size and composition of lipid rafts could change in response to intra- or extracellular stimuli[12,13]; such changes may favor specific protein-protein interactions and activate corresponding signaling cascades[14,15].

Human stomatin is ubiquitously expressed; the highest expression levels are noted in adipose tissues, bone marrow and placenta (Supplementary Fig. 1). At the cell level, stomatin is associated with the plasma membrane and cytoplasmic vesicles, such as endosomes[16], lipid droplets (LDs)[17], and specialized endosomes/granules in hematopoietic cells[2]. In the placenta, stomatin plays an important role in trophoblast differentiation[18]; and in bone, stomatin promotes osteoclastogenesis[6].

The high amount of stomatin in adipose tissues raises the possibility that stomatin may play important roles in the development and function of adipocytes, and that dysfunction of stomatin-mediated regulations on adipogenesis may lead to lipid metabolisms-associated disorders, such as obesity.

Obesity, characterized by the increased mass or functions of adipose tissues as energy harvest and expenditure become imbalance[19,20], is strongly associated with metabolic diseases such as diabetes, cardiovascular diseases, non-alcoholic fatty liver diseases and some cancers[21–23]. Both genetic and behavioral attributors contribute to obesity[24,25]. The expansion of adipose depots, especially the white adipose tissues, are characterized by the increase in adipocyte size (hypertrophy), or by formation of new adipocytes from the precursor cells (hyperplasia)[26–28]. In the presence of excessive energy, mature adipocytes increase in cell size and undergo cellular hypertrophy to store the surplus fat[29]. Hypertrophic adipocytes are characterized by excessive growth of LDs; the resulting unilocular LD may occupy more than 90% of the cell volume[30]. The hypertrophic adipocytes are responsible for dysfunction of lipid homeostasis, along with pathological consequences[31]; while adipocyte hyperplasia plays a role in preventing hypertrophy development and further maintaining the normal function of adipose tissue[32]. Approaches aimed at increasing adipogenesis or adipogenic differentiation, over adipocyte hypertrophy are now regarded as means to treat metabolic diseases. Notably, adipocyte expansion through adipogenesis could mitigate the negative metabolic effects of obesity[33].

In this work, we used the adipocyte-like cell model and gene manipulated animals to address the roles of stomatin in adipocyte differentiation and functions, focusing on stomatin-mediated modulations on signal transduction processes during adipogenesis, and on the cell biology of LDs growth and fatty acid uptake.

## Results

**Expressions of Stomatin increased during adipogenesis.** A cell model for investigating adipogenic differentiation has been established[34]. Treating murine 3T3-L1 fibroblasts with MDI cocktail for three days (Day 0 ~ Day 3), followed by insulin treatment (Day 3 ~ Day 7), caused these cells to differentiate into adipocyte-like cells, evidenced by intracellular lipid accumulation that could be visualized and quantified after Oil Red O staining (Fig. 1a). Expressions of stomatin and major adipogenic genes, such as PPARγ, C/EBPα, and Perilipin, progressively increased during this process (Fig. 1b). On Day 7, adipocyte-like cells were noted to contain large lipid droplets (LD) under DIC microscopy (Fig. 1c). Immunofluorescence staining revealed subcellular distributions of stomatin. In addition to puncta staining in the cytosol, stomatin proteins were noted on the plasma membranes (arrows), as well as surfaces of LDs (inset, Fig. 1c), where they partially colocalized with perilipin proteins when examined under STED microscopy (Fig. 1d). When LDs were isolated from adipocyte-like cells, stomatin was present in the LD fraction, together with the known LD-associated protein perilipin (Fig. 1e).

We used immunoprecipitation assays (IP) to identify proteins that bind to stomatin (Supplementary Fig. 2). A total of 885 candidates were recognized using Mass spectrometry; they could be grouped into six biological processes, including lipid transport, lipid homeostasis, lipid biosynthetic process, cholesterol transport, cholesterol homeostasis, and cholesterol biosynthetic process, indicating that stomatin were involved in lipid metabolisms. Among these "stomatin-associated proteins", perilipin-1 has been shown to participated in lipid catabolic process and unilocular lipid droplet formation[35].

**High level of stomatin enhances LD enlargement.** Human stomatin gene (hSTOM) was over-expressed in murine 3T3-L1 cells. These cells were induced to differentiate into adipocyte-like cells. We found that excessive exogenous stomatin proteins did not significantly affect adipogenic differentiation. The degrees of lipid accumulation (Fig. 2a) and expression levels of adipogenic genes (Fig. 2b) in hSTOM-over-expressing cells were comparative to the control cells.

By measuring the sizes of individual LDs, we noticed that there were more large-sized LDs in cells over-expressing hSTOM (gray bars, Fig. 2c) than in the control cells (black bars). A large LD could form by fusion of two small LDs. A time-lapse recording demonstrated such a LD-LD fusion event in an adipocyte-like cell over-expressing stomatin (Fig. 2d and Supplementary Movie 1).

Alternatively, a large LD could result from progressive "filling" of lipid content into such a LD by smaller LD vesicles; some of them were indiscernible under light microscopy. We employed fluorescence recovery after photobleaching (FRAP) experiments to investigate this possibility. As shown in Fig. 2e, photobleaching fluorescent content of a LD resulted in a faster recovery of fluorescence in adipocyte-like cells over-expressing stomatin (green, top traces), comparing to a much less and slower fluorescence recovery by the control cells (blue, bottom traces).

LD-LD fusion events were measured in vitro (Fig. 2f, g). LDs were isolated from cells whose LD were loaded with either green Bodipy-FL-$C_{16}$ or red Bodipy-558/568-$C_{12}$ fluorescently labeled fatty acid analogs (arrowheads, Fig. 2g). Some LDs, when mixed in vitro, appeared to undergo fusion, resulted in formation of

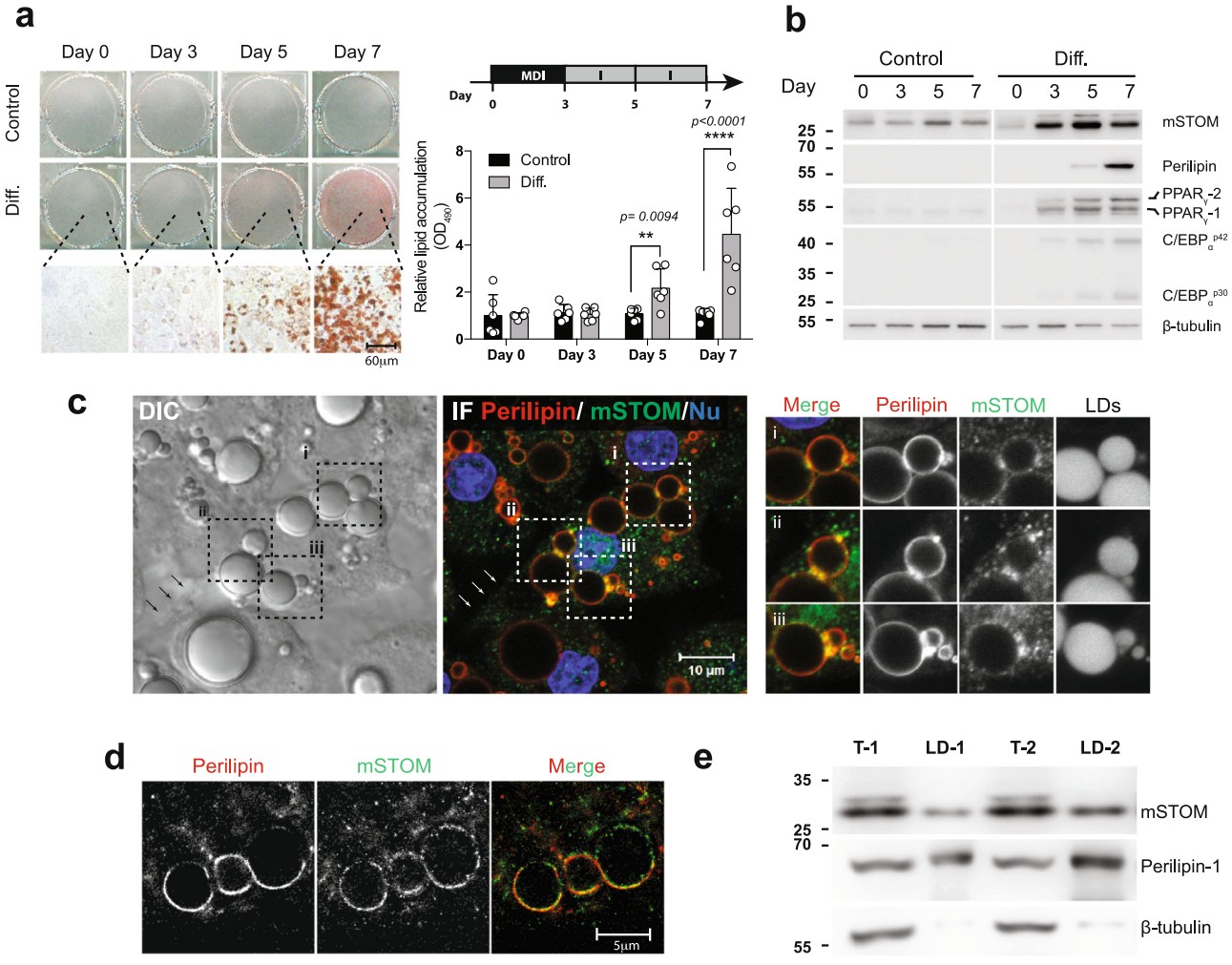

**Fig. 1 Stomatin expressions increased during adipogenesis. a** Mouse 3T3-L1 fibroblasts were treated with MDI induction medium (Diff.) or control vehicle. On Day 0, 3, 5, 7 after induction, cells were stained with Oil Red O and observed under microscopy. Bar = 60 μm. The amounts of lipid accumulation for individual culture plates were quantified by measuring absorbance at 490 nm (OD$_{490}$). Mean ± s.d. for six independent experiments. **$P < 0.01$ and ****$P < 0.0001$ by two-way ANOVA analysis. **b** On the day after induction as indicated, Western blotting analyses revealed increased expressions of endogenous mouse stomatin (mSTOM), perilipin, PPAR$_\gamma$ and C/EBP$_\alpha$. **c** Subcellular distributions of stomatin and perilipin in Day 7 adipocyte-like cells were observed by differential interference contrast (DIC) and immunofluorescence (IF) microscopy using a confocal microscope. Cell nuclei (Nu) were revealed by Hoechst stain. Stomatin proteins were present on the plasma membranes (arrows) and surfaces of lipid droplet (LD) (inset i-iii). Bar = 10 μm. **d** STED microscopy showed that stomatin and perilipin were partially colocalized on the LD surfaces. Scale bar = 5 μm. **e** LDs were isolated and subjected to immunoblotting analyses. Stomatin and perilipin proteins, but not β-tubulin, were identified in the LD fractions (LD-1 & LD-2). Source data are provided as a Source data file. STED: stimulated emission depletion; T-1 & T-2: total cell lysates.

yellow LD (arrows, Fig. 2g) as they contained both fluorescent fatty acid analogs. The degrees of LD-LD fusion could be quantified by calculating % fused LDs. Since stomatin could be excreted to the culture medium[6], we noticed that adding conditioned medium obtained from cells over-expressing stomatin (CM-hSTOM) resulted in increase of % fused LDs, compared to the conditioned medium harvested from the control cells (CM-Ctl). Interesting, depletion of stomatin by treating the conditioned medium with anti-stomatin antibodies effectively inhibited such fusion-promoting activity (Fig. 2g).

**High level of STOM facilitates fatty acid uptake.** Among 885 "stomatin-interacting proteins" identified (Supplementary Fig. 2), 183 proteins (about 21%) were present on plasma membranes (PM). Two such stomatin-associated PM proteins, CD36 and caveolin-1 (CAV1), played important roles in lipid storage[36]. The molecular interactions between stomatin and CD36 were demonstrated by immuno-precipitation experiments (Fig. 3a).

Fluorescently labeled palmitic acid analogs Bodipy-FL-C$_{16}$, when added to the culture medium, were internalized by 3T3-L1 adipocyte-like cells. The uptake process could be continuously monitored by measuring the increase of intracellular fluorescence (Fig. 3b). Adipocyte-like cells over-expressing stomatin exhibited more and faster fatty acid uptake (green, upper traces) than the control cells (blue, second traces). Such an effect was not caused by the diffusion. No significant change could be found in the Bodipy-FL uptaking efficiency between the over-expressing stomatin (gray traces) and control adipocyte-like cells (black traces).

**STOM recruits to the plasma membrane and interact with CD36.** Stomatin not only promoted fatty acid uptake, but also redistributed from LD surfaces to the plasma membranes during this internalization process. As shown in Fig. 3c, when treating cells with BSA as a control, stomatin proteins were present in high abundance surrounding LDs in about half of the adipocyte-like cells examined. However, when treating these cells with BSA-conjugated

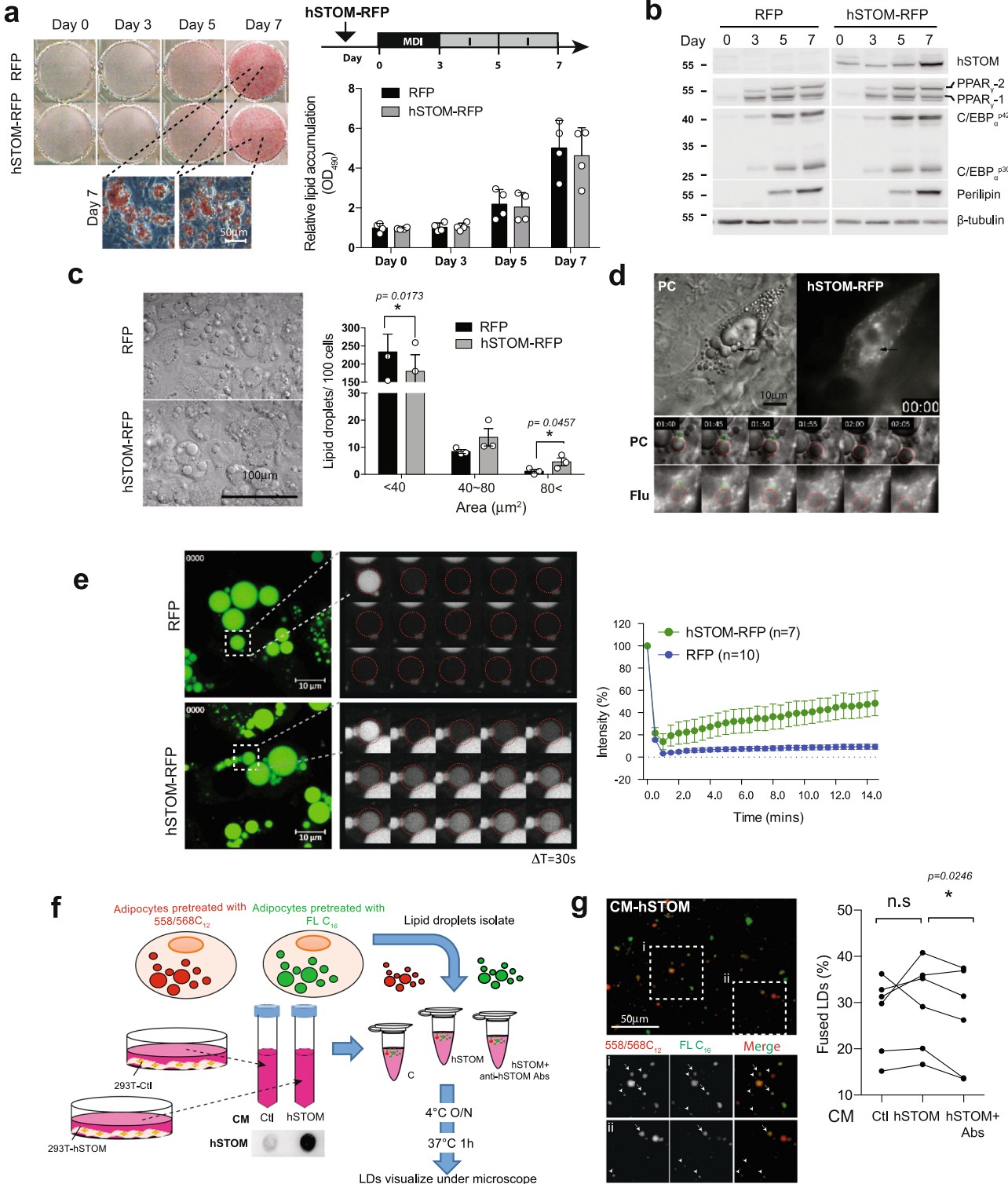

palmitic acid (PA) for 15 min, most of the LD-associated stomatin proteins seemed to "dissociate" from LDs and appeared on the plasma membrane or became dispersed in the cytosol.

The amount of membrane proteins located on the cell surface could be labeled using surface biotinylation methods and separately quantified from the proteins present inside the cell. As shown in Fig. 3d, the biotinylated portions on the surface (surface), relative to the total protein (input) of stomatin, CAV-1, CD36, and ATP1A1 were calculated. Actin proteins that present

only inside the cell were used as a control. We found that the surface portions of CAV-1, CD36, and ATP1A1 remained unchanged when treating cells with PA compared to BSA treatments. In contrast, the surface portion of stomatin increased when treating adipocyte-like cells with PA than with BSA (arrow). Such an increase could be due to PA-uptake triggered recruitment of stomatin from LD or other intracellular compartments to the plasma membranes that facilitated internalization of lipid ingredients from the extracellular environment.

**Fig. 2 Stomatin over-expression promoted LD growth and fusion in adipocyte-like cells. a** Mouse 3T3-L1 cells over-expressing human stomatin conjugated with RFP (hSTOM-RFP), or RFP as a control, were induced to differentiate. Lipid contents of the resulting adipocyte-like cells were stained and quantified. Mean ± s.d. for four experiments. Bar = 50 μm. **b** The protein of hSTOM, PPARγ, C/EBPα, and perilipin were examined by Western blotting. **c** The numbers of LDs were counted and plotted as a function of LD sizes by the histogram. Large LDs (>80 μm²) were more frequently found in hSTOM-RFP cells than control cells which contained mostly small (<40 μm²) LDs. Mean ± s.e.m. for three experiments. *P < 0.05 by two-sided paired t-test. Bar = 100 μm. **d** The time-lapse recording of a representative LD-LD fusion event (black arrow) when a small LD (circled green) fused with a large LD (circled red). Image tiles in hr:min. Bar = 10 μm. **e** FRAP experiments were done on cells transfected with hSTOM-RFP or RFP. After pre-treating cells with Bodipy-FL-C₁₂, fluorescence of a LD was photobleached (dashed box). Recovery of fluorescence, quantified as % of original intensity, was recorded at a 30-s interval. Mean ± s.e.m of LD. for three experiments. Bar = 10 μm. **f** In vitro assay for LD fusions. LDs isolated from adipocyte-like cells were pre-treated with either Bodipy-FL-C₁₆ or Bodipy-558/568-C₁₂. Two LD suspension individually labeled red and green, were mixed in vitro in the presence of conditioned-medium (CM) prepared from 293 T cells over-expressing hSTOM-Flag (hSTOM), or empty control (Ctl). Excreted stomatin in CM was depleted by anti-hSTOM antibodies. **g** Most isolated LDs exhibited green- or red-fluorescence (arrowheads) alone; some LDs, however, contained both (arrows) as a result of LD fusions. Stomatin-containing CM increased LD fusion, while depletion of stomatin (hSTOM+Ab) inhibited fusion. Data represent six experiments. *P < 0.05 by two-sided paired student t-test. Source data are provided as a Source data file.

Molecular interactions between stomatin and other LD-associated proteins, such as CD36 or CAV-1, were further analyzed in situ using proximity ligation assays (PLA). As shown in Fig. 3e, when treating 3T3-L1 adipocyte-like cells with PA, we noted significant increases of PLA + loci that indicated molecular binding between stomatin and CD36 along the plasma membrane (arrowheads). Withdrawal of PA treatments reversed such stomatin-CD36 interactions. In contrast, PA treatments did not affect interactions between stomatin and CAV-1 (Fig. 3f).

**Function of mutant STOM on adipocyte-like cells.** Stomatin contains several functional domains. In addition to the wild type, C-terminal truncated mutant (ΔC-hSTOM, 1-263aa) and N-terminal truncated mutant (ΔN-hSTOM, 54-288aa) were constructed (Supplementary Fig. 3a). Over-expression of all these constructs resulted in increase of large LDs (Supplementary Fig. 3b) in adipocyte-like cells; however, the subcellular distribution (Supplementary Fig. 3a) and fatty acid uptake function of C-terminal truncated mutant was comparable to the wild type (Supplementary Fig. 3c). In contrast, the ΔN-hSTOM displayed puncta-like signals inside the cells and slightly reduced the degree of FA uptake compared to the wild type (Supplementary Fig. 3c). All mutant proteins, like the wild type proteins, could bind to CD36 (Supplementary Fig. 3d). The content and distribution of free cholesterol inside the cell measured by filipin staining (Supplementary Fig. 4a) and internalization of extracellular cholesterol into the cell quantified by uptake of fluorescently-labeled CholEsteryl (CholEsteryl Bodipy 542/563 C₁₁) added to the culture medium (Supplementary Fig. 4b), were not affected by over-expression of wild type or mutant stomatin. Knockdown of stomatin, on the other hand, decreased the free cholesterol content (Supplementary Fig. 4c) and down-regulated cholesterol uptake from outside of the cell (Supplementary Fig. 4d).

The CAV-1 distribution inside the cell was also measured by immunofluorescence staining with the CAV-1 antibody (Supplementary Fig. 4e). Knockdown of stomatin (shSTOM) decreased the amount of CAV-1 on the plasma membranes compared to control (Ctl) cells.

**Elevated STOM causes obesity in mice fed with high-fat diet.** Effects of diet on stomatin expressions were studied. Wild-type mice (WT) were fed with either regular chow diet (CD) or high-fat diet (HFD) for 20 weeks. The fat pads were collected and their stomatin expressions were quantified by qPCR assay (Fig. 4a). We found that mRNA of mSTOM increased in both subcutaneous (SAT) and visceral (VAT) white adipose tissues in HFD-fed mice, compared to CD-fed mice, but such increase was less apparent in brown adipose tissue (BAT).

The STOM transgenic (STOM Tg) mice were generated by engineering human stomatin gene into the animal. These mice contained high amounts of hSTOM proteins in SAT (Fig. 4b), that were found mainly on the surface membranes of white adipocytes (Fig. 4c). STOM Tg and the control WT mice were fed with CD or HFD since aged 3-week for 20 weeks; they were weighed every week (Fig. 4d). Body weight gains were similar comparing STOM Tg with WT mice when fed with CD. However, when fed with HFD, mice with excessive stomatin gained more weight more rapidly than their WT littermates (Fig. 4d, e). After 20 weeks of HFD feeding, the body weights of STOM Tg mice were at least 17% higher than the WT mice. Whole-body composition measurements showed that gains of fat were more significant (an averaged increase of 32%) than lean, free fluid, or total water (Fig. 4f).

In vivo fatty acid uptake experiments were done by injecting BSA-emulsified Bodipy-FL-C₁₆ into the tail vein of mice, incubating for 15 min, then measuring the fluorescence signals in the white adipose tissue of the animal (Fig. 4g). We noticed that there was more fatty acid uptake to the adipose tissues in STOM Tg mice than in WT mice (Fig. 4g).

After 20-week HFD feeding, mice having up-regulated stomatin showed increased mass of SAT and BAT than WT mice, while their VAT tissues remained relatively unchanged (Fig. 5a). Histological analyses showed that adipocytes of SAT in HFD-fed STOM Tg mice appeared hypertrophic and larger in size than those of WT littermates (Fig. 5b). However, Western blotting showed that enzymes for lipolysis, including perilipin and hormone-sensitive lipase (HSL) and its various serine-phosphorylated forms, were similar in hSTOM-overexpressed 3T3-L1 adipocytes compared to the control (Supplementary Fig. 5a).

**HFD-fed STOM Tg mice exhibited metabolic impairment.** The observed obesity was unlikely due to altered energy expenditure or energy substrate selection, as evidenced by similar respiratory exchange rates comparing HFD-fed STOM Tg mice with their littermate controls (Supplementary Fig. 5b). The calculated heat production during the light-dark cycle was slightly higher in HFD-fed STOM Tg mice than in controls, thus could not be accounted for the observed obesity phenotype (Supplementary Fig. 5c). Biochemistry study of the blood showed that although total cholesterol (TCHO) in HFD-fed STOM Tg appeared slightly higher than HFD-fed WT mice, their triglycerides (TG) levels were comparable (Supplementary Fig. 5d).

As shown in Fig. 5c, although fasting blood glucose levels were similar between STOM Tg and WT mice, plasma insulin and insulin resistance measured by HOMA-IR experiments were significantly higher, and glucose tolerance measured by

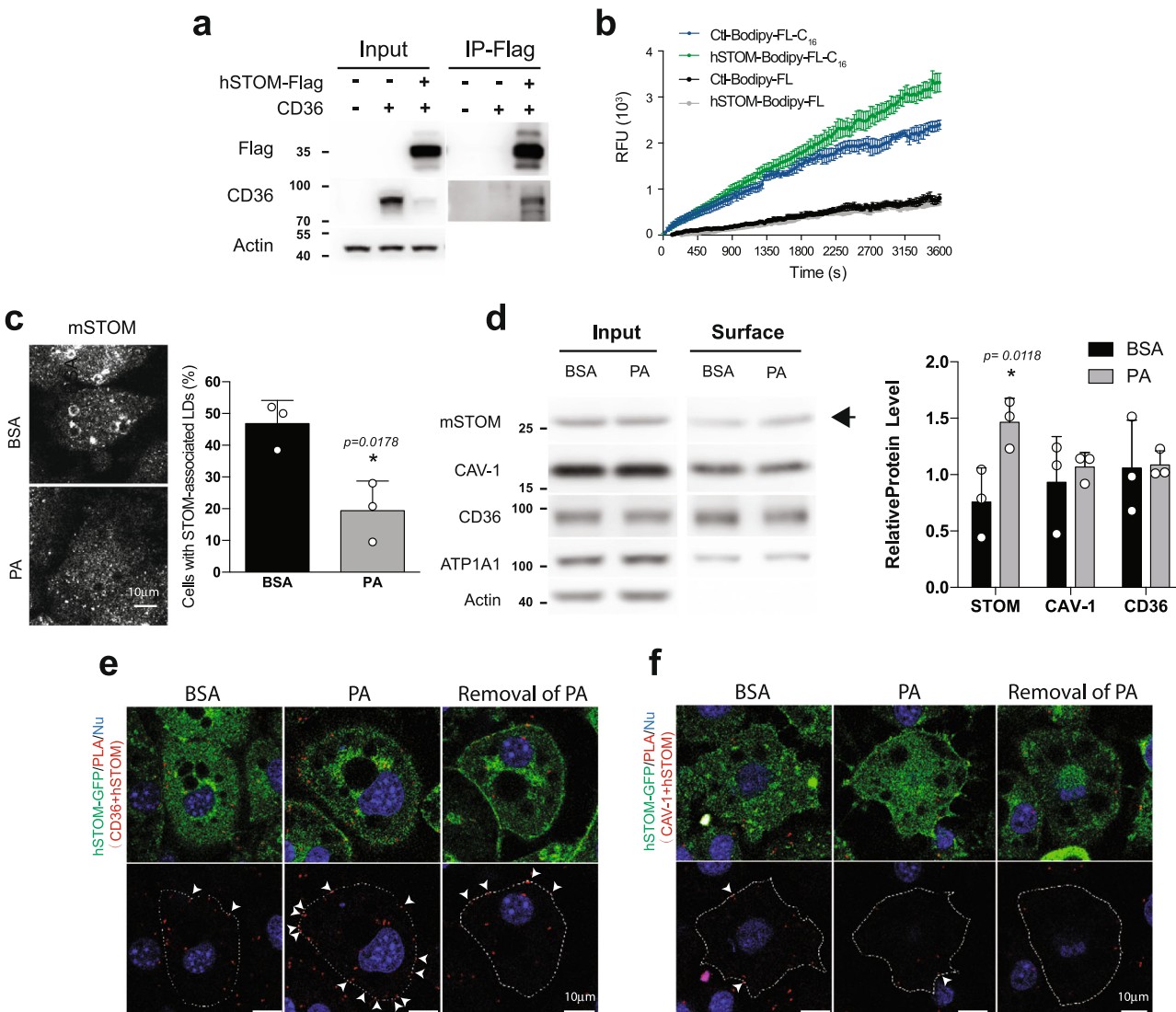

**Fig. 3 Stomatin interacted with CD36 and facilitated fatty acid uptake. a** Immunoprecipitation (IP) assays revealed molecular interactions between stomatin and CD 36 in vitro. **b** Fatty acid uptake assays were performed to measure internalization of lipid ingredients by adipocyte-like cells from the extracellular environment. Fluorescently-labeled fatty acid (Bodipy-FL-$C_{16}$) or fluorescent tag alone (Bodipy-FL) was added to the culture medium of adipocyte-like cells expressing exogenous stomatin (hSTOM), or control vector (Ctl). The degrees of Bodipy-FL-$C_{16}$ and Bodipy-FL internalization were measured by intracellular accumulations of fluorescence over time. Mean ± s.e.m. for at $n = 3$ for Bodipy-FL-$C_{16}$ $n = 6$ for Bodipy-FL uptake independent cells. **c** Immunofluorescence staining demonstrated that in cells treated with BSA, about half of the cell population contained stomatin-associated LDs. Treating cells with palmitic acid (PA) resulted in dissociation of STOM from LDs. Mean ± s.d. for three independent experiments. *$P < 0.05$ by two-sided unpaired $t$-test. Bar = 10 μm. **d** Surface biotinylation followed by Western blotting analyses of the biotinylated proteins were performed to quantify the surface portion (Surface) of a membrane protein, including stomatin, CD36, caveolin-1 (CAV1), or ATP1A1, relative to the total amount of individual proteins (Input). Actin proteins could not be labeled by surface biotinylation method and therefore contained no surface portion. Note increase of surface portion of stomatin after PA treatments (arrow) compared to the BSA control. In contrast, the surface portion of CD36, CAV1 remained unchanged after PA treatments. Mean ± s.d. for three independent experiments. *$P < 0.05$ by two-way ANOVA analysis. **e, f** Proximity ligation assays (PLA) were performed in 3T3-L1 adipocyte-like cells over-expressing exogenous human stomatin fused with GFP (hSTOM-GFP). Duolink® PLA reactions occurred when PLA probes for CD36 **e** or for CAV1 **f** were in closed proximity with hSTOM-GFP. Addition of connector oligos joined the PLA probes, resulting in in situ proximity ligation and formation of amplicon (arrowheads). **e** PA treatments promoted interactions between CD36 and stomatin on the plasma membrane, as evidence by the increased amplicons along cell surface compared with the control BSA treatments; removal of PA reversed such an increase. On the other hand, **f** PA treatments did not affect interactions between calveolin-1 and stomatin. Bar = 10 μm. Source data are provided as a Source data file.

intraperitoneal glucose tolerance test (IPTGG) was more intolerable in HFD-fed *STOM* Tg than the control mice.

Obesity is often associated with ectopic fat accumulation in the liver. Indeed, HFD-fed *STOM* Tg mice exhibited larger liver mass (hepatomegaly), and macro- and microvesicular steatosis. These phenotypes were related to impaired liver functions, as evidenced by elevated serum levels of GPT and GOP (Fig. 5d).

**Depleted STOM blocks the adipogenesis and LD enlargement.** Knockdown of stomatin gene was done by the short hairpin RNA (shRNA) method. Two shRNAs designed to target different sites of murine stomatin gene were separately packaged into lentiviral particles and introduced into 3T3-L1cells, resulting in shSTOM-1 and shSTOM-2 cells. Both shRNAs could effectively down-regulate stomatin expressions. Knockdown of stomatin was noted to inhibit

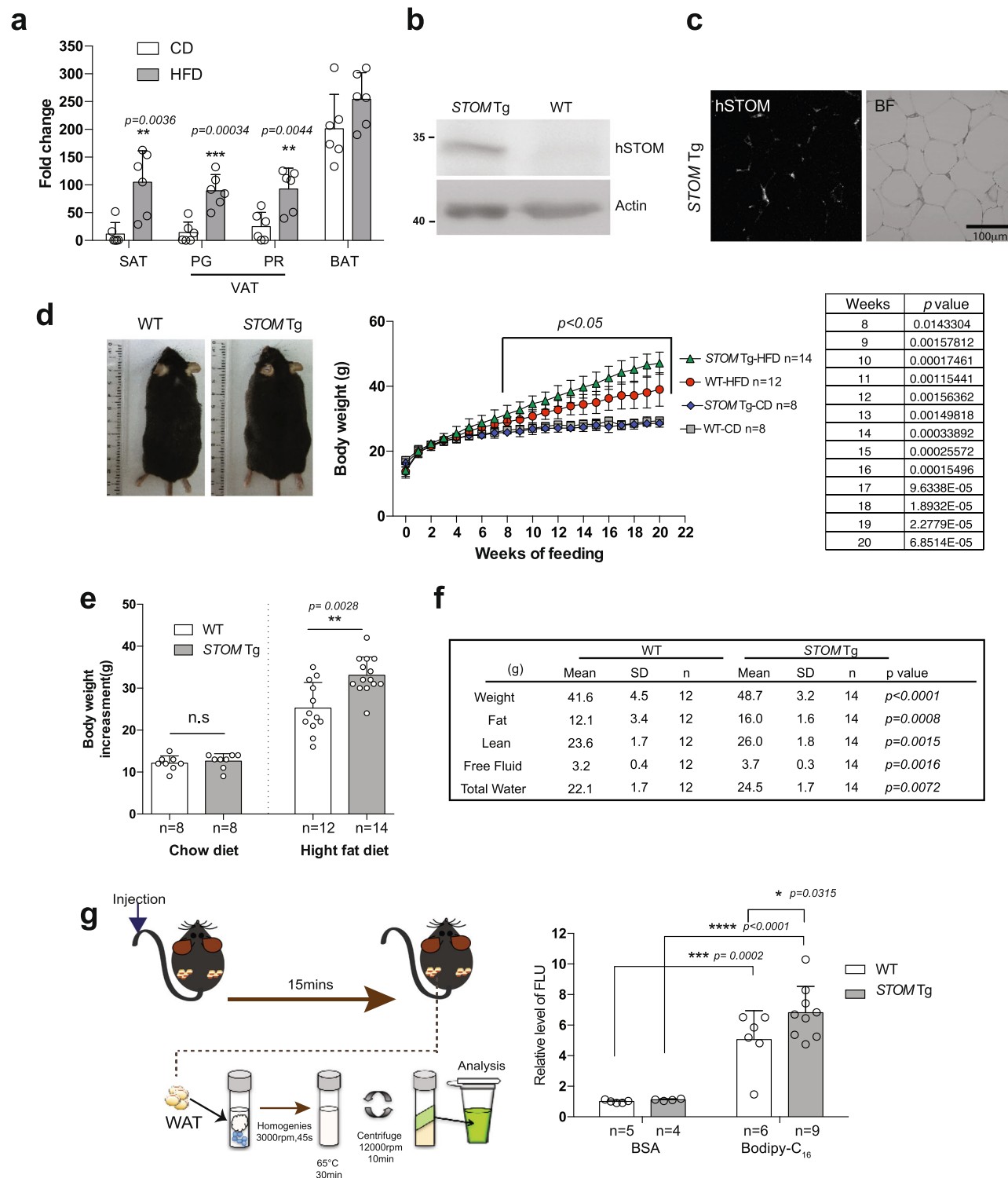

adipogenesis, as evidenced by lack of lipid accumulation after induction of differentiation (Fig. 6a). Expressions of genes involved in adipocytic differentiation, such as PPARγ and C/EBPα, were decreased by stomatin knockdown (Fig. 6b). Down-regulation of STOM also inhibited LD maturation and growth. Histogram analyses showed that there were more small-LD and less large-LD in STOM knockdown cells than the controls (Fig. 6c).

To knockdown stomatin locally at fat tissues instead of whole body, we injected AAV virus carrying shSTOM/GFP or GFP at SAT fat pad of mice (Supplementary Fig. 6a). The targeted SAT

tissues were dissected and analyzed for their masses and histological sections (Supplementary Fig. 6b). We found that after HFD-feeding, the mass of AAV-infected SAT was slightly lower than the control SAT (Supplementary Fig. 6c). Using immunostaining of Na-K-ATPase for cell size measurements, we also noticed that the cells at the local shSTOM/GFP injected tissue were smaller than the cells on the tissues injected with control GFP vector (Supplementary Fig. 6d).

Differential expression analyses were performed comparing stomatin-knockdown and control 3T3-L1 adipocyte-like cells. As

**Fig. 4 Stomatin transgenic mice fed with high-fat diet were more obese than the control mice. a** Real-time qPCR analyses validated increased stomatin gene expressions in SAT, VAT, including Perigonadal fat pad (PG) and Perirenal fat pad (PR), and BAT after regular CD or HFD feeding. Mean ± s.d. for six mice. **$P < 0.01$ and ***$P < 0.001$, by multiple unpaired $t$ test. **b** Western blotting revealed high expressions of human stomatin (hSTOM) proteins in stomatin transgenic mice (*STOM* Tg). **c** The exogenous hSTOM proteins were present mainly on the plasma membranes of adipocytes in fat tissues of *STOM* Tg mice. **d** Body weight changes of *STOM* Tg and wild type (WT) mice during CD or HFD feeding for 20 weeks. Mean ± s.d. of mice, *$P < 0.05$ analyzed by multiple unpaired $t$ tests. Representative photos of the mice are shown. **e** Body weight increments after 20-week feeding were measured. Each dot represents one mouse. While no difference was noticed in animals fed with CD, body weight gains were significantly higher in HFD-fed *STOM* Tg, compared to HFD-fed WT mice. Mean ± s.d. is shown. n.s = non-significant. **$P < 0.01$ by two-sided unpaired $t$-test. **f** The mass of whole body, fat, lean, free fluid, and total water were calculated by body composition analyzer for HFD-fed *STOM* Tg and HFD-fed WT mice. $P$ values were examined by multiple unpaired $t$ test. **g** Fatty acid uptake was measured in vivo. Fluorescently-labeled fatty acid Bodipy-FL-C$_{16}$, or BSA, were injected into tail vein of *STOM* Tg or WT mice. After 15 min, lipid portions of white adipose tissue (WAT) from the animal were extracted and their fluorescence signals that represented lipid uptake were quantified. Data shown are fold changes of fluorescence intensity using BSA injected to WT mice as the reference. Each dot represents one mouse. Mean ± s.d. for mice from two independent experiments. *$P < 0.05$, ***$P < 0.001$, and ****$P < 0.0001$ by two-way ANOVA. Source data are provided as a Source data file.

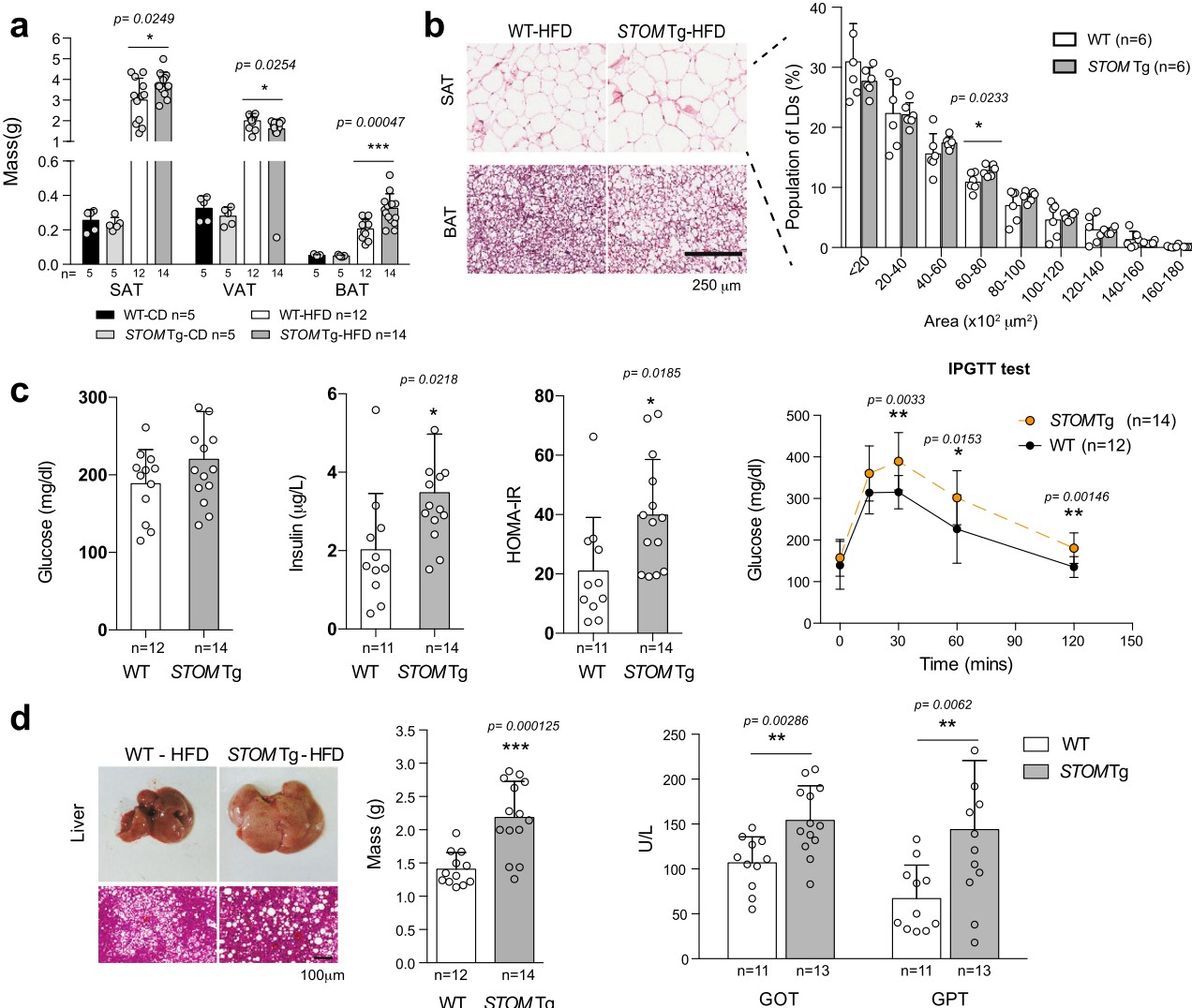

**Fig. 5 Stomatin transgenic mice fed with high-fat diet exhibited adipocyte hypertrophy and metabolism disorders. a** The weights of SAT, VAT and BAT in stomatin transgenic mice *STOM* Tg or WT mice, fed with regular CD or HFD for 20 weeks were measured. **b** Representative H&E-stained histopathological sections of SAT, and BAT from HFD-fed *STOM* Tg or HFD-fed WT mice. Histogram analyses of sizes of adipocytes are shown. Bar = 250 μm. **c** After HFD-feeding for 20 weeks, the animal's fasting glucose, serum insulin, homeostatic model assessment of insulin resistance (HOMA-IR), and intraperitoneal glucose tolerance test (IPGTT) were examined. **d** The mass of liver of *STOM* Tg or WT mice, fed with HFD for 20 weeks, were weighed. Fatty liver changes revealed by histopathological sections and high level of serum glutamate oxaloacetate transaminase (GOT) and glutamate pyruvate transaminase (GPT) indicated impaired hepatic functions in *STOM* Tg mice, compared to WT mice. Bar = 100 μm. **a–d** Each dot represents one mouse. Mean ± s.d. is shown. n.s = non-significant. *$P < 0.05$, **$P < 0.01$, and ***$P < 0.001$ by multiple unpaired $t$-test. Source data are provided as a Source data file.

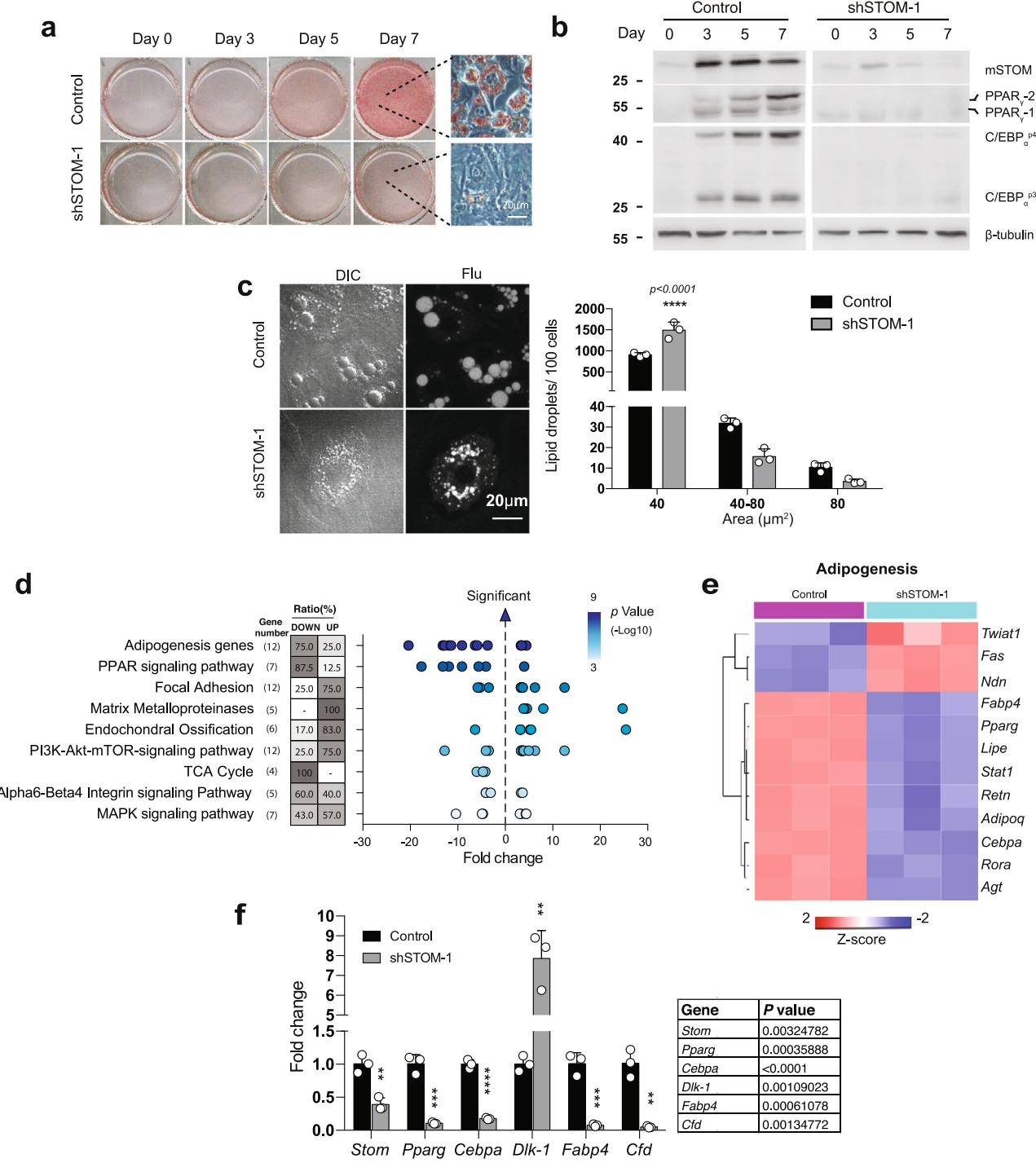

shown in Supplementary Fig. 7a, a total of 1478 annotated coding genes were identified from the transcriptomes by a stringent threshold of $P < 0.05$ and FDR $P < 0.001$. Among them, 379 transcripts were of significant difference between STOM-deficient and control cells (fold change ≥3, or ≤ −3), consisting 185 up-regulated and 194 down-regulated genes. The global view of these genes was constructed by hierarchical clustering to characterize changes across six samples (Supplementary Fig. 7b). These genes were mapped onto 128 Wikipathways using Transcriptome Analysis Console (TAC). As shown by the top-ranking enriched pathways (Fig. 6d), adipogenesis genes were the most profoundly inhibited gene groups, followed by the PPAR

signaling pathway. In the "adipogenesis genes" pathway, 12 genes involved in this pathway showed significant changes; 9 of them (75%) were down-regulated and 3 (25%) were up-regulated (Fig. 6e). To further validate the microarray results, we performed qPCR experiments on individual genes, focusing on adipogenesis-related genes *Pparg, Cebpa, Dlk-1, Fabp4, and Cfd* genes (Fig. 6f); all of them were down-regulated, except for *Dlk-1*.

To study whether stomatin deficiency influences HFD-induced obesity in mice, a stomatin-knockout mouse ($Stom^{-/-}$) was generated (Supplementary Fig. 8a). When fed with HFD, $Stom^{-/-}$ mice exhibited a slightly less weight gain than $Stom^{+/-}$ mice in the first 4 weeks (Supplementary Fig. 8b), but gradually caught up

**Fig. 6 Decreased stomatin expression affected adipogenesis and inhibited lipid droplet growth. a** Stomatin knockdown was done by transduction of shRNA of *Stomatin* gene into 3T3-L1 cells generating shSTOM-1 cells or shSTOM-2 cells. After induction of adipogenic differentiation, shSTOM-1 and control cells were stained with Oil Red O. Cells after 7-day induction are shown using high-power microscopy. Bar = 20 μm. **b** On the day after induction as indicated, Western blotting analyses revealed decreased expression of mouse stomatin (mSTOM) by knockdown. Adipogenic proteins, including PPARγ, and C/EBPα were also down-regulated. **c** On Day 7 of adipogenic differentiation, adipocyte-like shSTOM-1 cells were stained with Bodipy-FL and observed under differential interference contrast (DIC) and fluorescence (Flu) microscopy. The sizes of lipid droplets (LDs) were measured, and their numbers per 100 adipocytes were calculated. Histogram analyses showed more small LDs (<40 μm$^2$) and fewer large (>80 μm$^2$) LDs in shSTOM-1 cells than the control. Mean ± s.d. for three independent experiments. ****$P$ < 0.0001 by two-way ANOVA. Bar = 20 μm. **d** Transcriptome analyses of adipocyte-like shSTOM-1 and control cells after induction for 7 days. From the data of microarray assays, scatter plots revealed enriched Wiki pathways in shSTOM-1 cells, compared to the control. For a given pathway, ratio of gene number being up- or down-regulated in that pathway were determined, and plotted as function of fold change. Each dot represents one gene. The color of the dots represents the range of $P$-values related to the indicated pathway are provided by the Transcriptome Analysis Console (TAC). **e** Heat map of adipogenesis gene obtained from the enrichment-based cluster analysis of the Wiki pathway. **f** Real-time qPCR analyses to validate the changed genes revealed by microarray assays. Data shown are fold changes relative to *Nono* mRNA level. Mean ± s.d. for three independent experiments. **$P$ < 0.01, ***$P$ < 0.001, ****$P$ < 0.0001 by multiple unpaired *t*-test. Source data are provided as a Source data file.

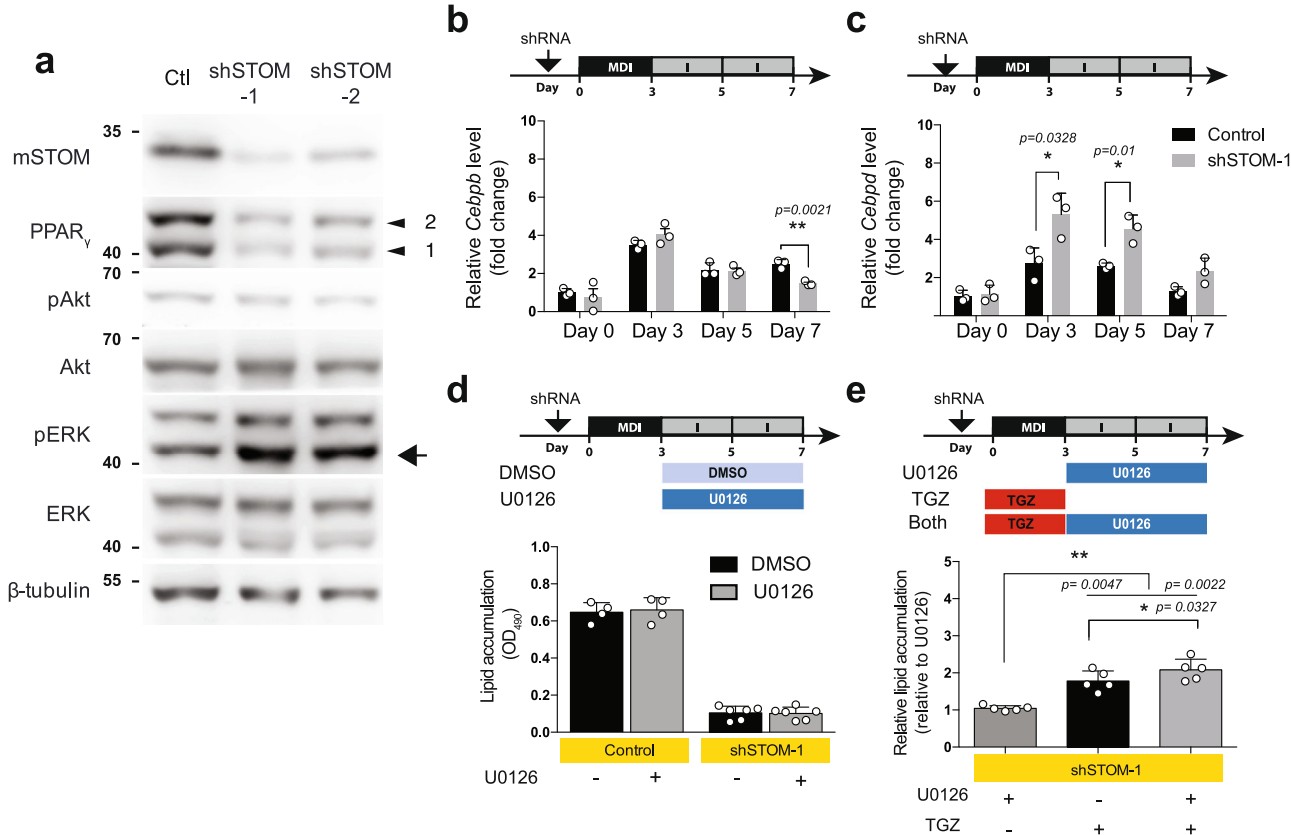

**Fig. 7 Knockdown of stomatin activated ERK-pathway. a** shRNA was transduced into 3T3-L1 cells to knockdown stomatin, generating shSTOM-1 and shSTOM-2 cells. After induction of differentiation, Western blotting analyses showed that although Akt pathway remained unchanged, the ERK pathway was activated, as evidenced by increased pERK (arrow). **b, c** Expressions of early adipogenic genes, *Cebpb* and *Cebpd*, were examined at the mRNA level by qPCR. **b** Expressions of *Cebpb* increased in the first three days of adipogenic differentiation, and subsequently decreased. Knockdown of stomatin (exemplified by shSTOM-1 cells) did not significantly influence this pattern. **c** Expressions of *Cebpd* gene also exhibited a transient early increase then declined; however, knockdown of stomatin appeared to significantly increase and maintain *Cebpd* expression at a relatively higher level than the control. **b, c** The house-keeping *Nono* gene was used as the reference for qPCR experiments. *$P$ < 0.05, and **$P$ < 0.01 by multiple unpaired *t*-test. **d** Levels of lipid accumulation by adipocyte-like 3T3-L1 cells were quantified by measuring OD$_{490}$ after Oil Red O staining. Treating shSTOM-1 cells with 10 μM U0126, an ERK pathway inhibitor, from Day 3 to Day 7, did not reverse the inhibition of lipid accumulation caused by stomatin knockdown. Mean ± s.d. for n = 4 (Control) and n = 6 (shSTOM-1) independent experiments. **e** On the other hand, treating shSTOM-1cells with TGZ, a PPARγ activator, in the first three days of adipogenic differentiation, was able to partially recover the knockdown-caused lipid accumulation deficit; treating shSTOM-1cells with both TGZ and U0126 could further increase lipid accumulation. Mean ± s.d. for five independent experiments. *$P$ < 0.05, and **$P$ < 0.01 by one-way ANOVA. Source data are provided as a Source data file.

with the body weights of $Stom^{+/-}$ mice. The body weight increments were not significantly different comparing $Stom^{-/-}$ with $Stom^{+/-}$ mice fed with either CD or HFD (Supplementary Fig. 8c). The mass of SAT, VAT and BAT were also similar

(Supplementary Fig. 8d). Histological analyses only revealed slightly smaller adipocytes in fat pads of $Stom^{-/-}$ mice compared to $Stom^{+/-}$ littermates (Supplementary Fig. 8e). Expressions of adipogenic genes in HFD-fed $Stom^{-/-}$ mice also resembled those

of HFD-fed $Stom^{+/-}$ mice. (Supplementary Fig. 8f). Our results suggest the presence of compensation mechanisms in $Stom$–null mice for adipogenesis.

**STOM modulates adipogenesis through ERK pathway**. As shown in Fig. 7a, knockdown of stomatin resulted in a decrease of PPAR$_\gamma$. The Akt pathway appeared not affected by STOM knockdown, while the ERK pathway was activated, as evidenced by the increased phosphorylated ERK (arrow).

The adipogenesis genes, C/EBP$_\beta$ and C/EBP$_\delta$, are upstream of PPAR$_\gamma$ regulation. After induction of adipogenic differentiation, C/EBP$_\beta$ and C/EBP$_\delta$, exhibited a transient increase in the first three days, followed by a gradual decrease from Day 3 to Day 7 (black bars, Fig. 7b, c, respectively). In stomatin knockdown 3T3-L1 adipocyte-like cells, we found that stomatin deficiency caused a decrease of C/EBP$_\beta$ on Day 7 (Fig. 7b). In contrast, stomatin knockdown was able to maintain or increase C/EBP$_\delta$ expression to a much higher level than the control (Fig. 7c).

Activation of ERK pathway has been shown to inhibit PPAR$_\gamma$[37], and thereby prohibited adipocytic differentiation and adipogenesis. U0126 is a highly selective inhibitor for ERK. We treated shSTOM-1 adipocyte-like cells with U0126 under the notion that U0126 might mitigate the stomatin-knockdown effect on pERK activation. However, U0126 treatments did not reverse the lipid accumulation-deficit phenotype of hSTOM-1 cells (Fig. 7d). In contrast, treating shSTOM-1 cells with troglitazone (TGZ), a PPAR$_\gamma$ agonist, was able to partially rescue lipid accumulation-deficit; and interestingly, dual treatments of U0126 and TGZ could further promote lipid accumulation (Fig. 7e). These results suggest the presence of a currently unknown mechanism for stomatin to positively regulate PPAR$_\gamma$ and activate adipogenesis, independent of ERK (see below).

Functional inhibition of stomatin could be achieved by a pharmacological reagent OB-1[38] which, by interfering self-association of stomatin, inhibited stomatin's activity. $LD_{50}$ for OB-1 was first determined (Fig. 8a). Treating 3T3-L1 adipocyte-like cells with OB1 effectively reduced lipid accumulation in a dose-dependent manner (Fig. 8b). The sizes of LDs were also reduced (Fig. 8c) and levels of phospho-ERK were increased (Fig. 8d) by OB-1 treatments.

## Discussion

Based on results shown here, we propose working models for roles of stomatin in modulation of adipogenic differentiation (Fig. 9a) and in regulations of LD growth and fatty acid uptake in adipocytes (Fig. 9b).

As shown in Fig. 9a, an undifferentiated progenitor cell can be induced to differentiates into an immature adipocyte, then to a mature adipocyte. Our results demonstrate a transient increase of two early adipogenic differentiation genes[39], C/EBP$_\beta$ and C/EBP$_\delta$ during adipogenic differentiation. The rise of C/EBP$_\alpha$, however, occurs at a relatively later phase[39]. These adipogenic genes seem to converge to PPAR$_\gamma$, which serves as a master regulator for signaling pathways that lead to adipocytic differentiation[40]. Expressions of stomatin increase and maintain at a relatively high level appear crucial for the differentiation process, as down-regulation of stomatin can profoundly inhibit adipocytic differentiation at the cell level. However, only very mild phenotypes can be observed in stomatin-knockout mics[5], possibly due to compensation effects. Stomatin may negatively regulate C/EBP$_\delta$, as down-regulation of stomatin keeps C/EBP$_\delta$ at a high level, but does not seem to affect C/EBP$_\beta$. Mechanisms for stomatin's regulations and interactions with C/EBP$_\beta$, C/EBP$_\delta$, or C/EBP$_\alpha$ are still unclear.

pERK is a well-known negative regulator for PPAR$_\gamma$[37]. We show here that down-regulation of stomatin can increase pERK and activate ERK pathway. We reason that a continued presence of stomatin at the later phase of adipogenic differentiation can down-regulate pERK (Fig. 9a), and by doing so, activates PPAR$_\gamma$ pathway and promotes downstream adipocytic phenotype. However, as shown in Fig. 7d, direct activation of ERK pathway did not rescue adipogenesis deficit caused by stomatin knock-down, suggesting the presence of another regulatory role for stomatin to activates PPAR$_\gamma$ independent of pERK (Fig. 9a). These two mechanisms may work synergistically to initiate and further promote adipocytes differentiation.

Transcriptome and pathway analyses reveal that MAPK signaling is affected by stomatin (Fig. 6d and Supplementary Fig. 7). How can stomatin, a lipid rafts component, serve as a signaling regulator? This may be due to stomatin's activity in controlling the metabolisms and interactions of membrane receptor proteins residing within the lipid rafts. It has been shown that EGFR can serve as an upstream RTK to control many fundamental cell behaviors, including adipogenic differentiation, through a series of downstream signaling pathways, including RAS-RAF-MEK-ERK pathway or AKT-PI3K-mTOR pathway[41,42]. Metabolisms of EGFR on the cell surface can be regulated by lipid-raft mediated internalization, recycling processes and degradation of the membrane receptor[43–45]. Stomatin may cause changes of EGFR density on the plasma membrane affect the ERK activity, essential for adipogenic differentiation. Stomatin can also affect caveolae-mediated endocytosis as down-regulations of stomatin appeared to reduce the amount of CAV-1 on the cell membrane (Supplementary Fig. 4e).

Besides modulating metabolisms of surface receptors and signaling molecules, stomatin also plays a crucial role in controlling maturation, growth and function of LDs (Fig. 9b). LD is a universal cellular organelle responsible for lipid storage[46]. In mature adipocytes, LDs growth is characterized by vesicle enlargement and transformation from multilocular LDs to unilocular LDs[47] which are typically seen in adipose tissues. Stomatin is a major LD-associated protein[17]. We have previously shown that stomatin could promote membrane fusion by formation of potential fusion pores; other fusion facilitators could then be recruited to the contact site and initiate the fusion process[6,18]. As depicted in Fig. 9b, when two LDs contact with each other (a), stomatin and perilipin were mobilized to the contact site (b); such aggregation of molecular complexes then triggered LD fusion processes (c). Direct observations of LD fusion are hard to achieve due to either phototoxicity that hinder the fusion biology, or the fact that LDs undergoing fusion are submicron in size and are often indiscernible under light microscopy. However, the outcome of fusion, such as replenishing of LD contents, can be readily measured by FRAP experiments (Fig. 2e). As also been demonstrated in other studies, stomatin could help generate passage tunnels that allowed lipid exchange between contacted LDs[35,48]. Between a contacted LD-LD pair, lipid contents of the smaller LD could be transported to the larger LD due to internal pressure difference[35,48].

A mature adipocyte is capable of internalizing fatty acid of superfluous lipids from the extracellular environment. We demonstrated here that the amounts and rates of fatty acid uptake inside the cell were increased by high-level stomatin expressions both in vitro (Fig. 3b) and in vivo (Fig. 4g). Although some fatty acids can cross plasma membrane by passive diffusion[49], most fatty acid uptake is mediated by membrane-associated transporters; many of them reside and function in the lipid rafts, including CD36 and a variety of fatty-acid-binding proteins (FABPs). CD36, also known as fatty acid translocase (FAT), is an integral membrane protein found on the surface of many cell types in vertebrate animals. Long-chain fatty acid (LCFAs) can bind to

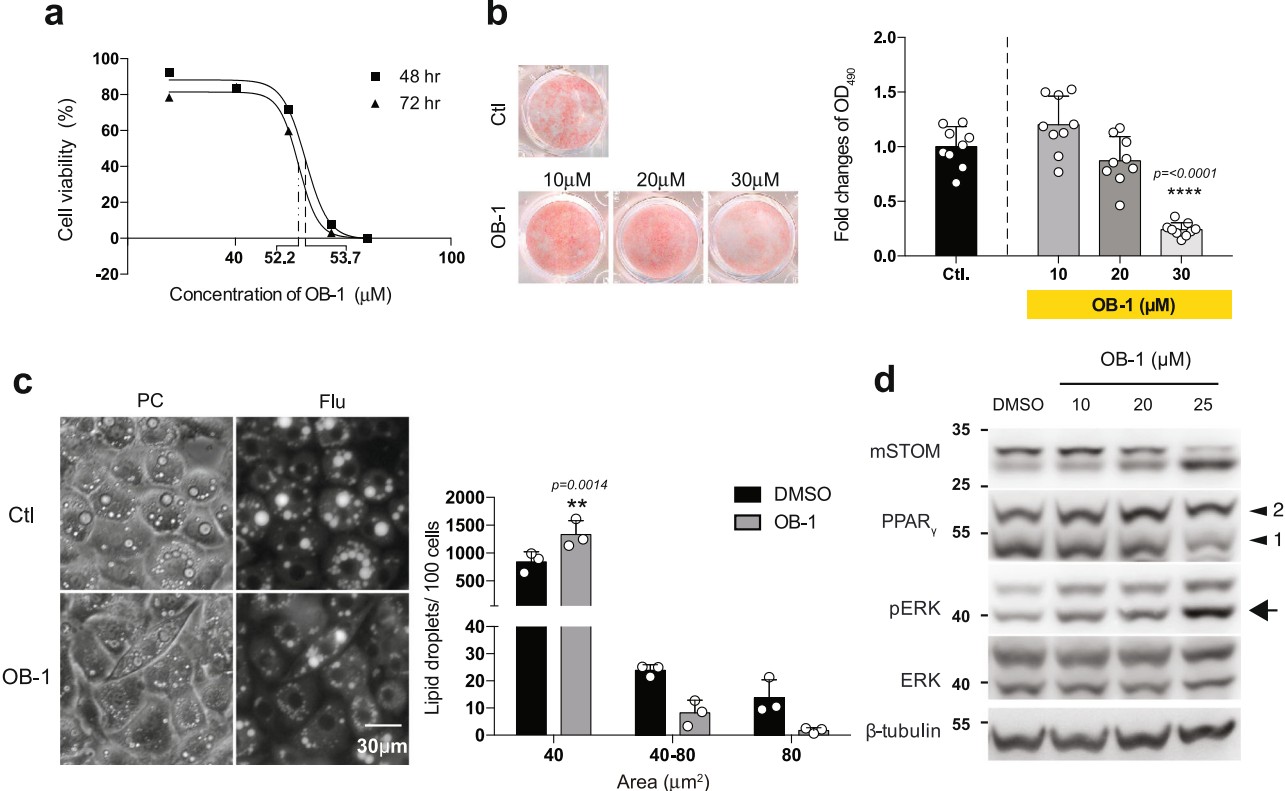

**Fig. 8 Stomatin inhibitor OB-1 inhibited adipogenesis and hindered lipid droplet growth. a** Cell viability after OB-1 treatments for 48 h or 72 h were tested by MTT assays. LC$_{50}$ for OB-1 were tested from three independent experiments. **b** Cultured plates of adipocyte-like cells were treated with or without OB-1 at concentrations indicated, stained with Oil Red O and subjected to OD$_{490}$ quantifications. Note OB-1 inhibited adipogenesis in a dose-dependent manner, compared to the control DMSO treatment. Each OB-1 treatment data was normalized to the corresponding DMSO vehicle control; fold changes are shown. Mean ± s.d. for three independent experiments ($n = 3$ for each experiment). ****$P < 0.0001$ by one-way ANOVA. **c** Adipocyte-like 3T3-L1 cells after 7-day induction were treated with 25 μM OB-1 or control vehicle; LDs of these cells were labeled by AdipoRed. The numbers and sizes of LDs were determined. Histogram analyses showed more small LDs (<40 μm$^2$) and less large LDs (>80 μm$^2$) after OB-1 treatments, compared to the control. Mean ± s.d. for three independent experiments. **$P < 0.01$ by two-way ANOVA. Bar = 30 μm. **d** Treating adipocyte-like cells with stomatin inhibitor OB-1 also resulted in activation of ERK pathway, evidenced by increased pERK (arrow). Source data are provided as a Source data file.

CD36. The resulting FAT/CD36 may partition into lipid rafts (1, Fig. 9b, see also ref. [50,51]). Exposure of adipocyte to LCFAs is noted to also relocate stomatin proteins from LDs to the plasma membrane, especially to the lipid rafts (2). In lipid rafts, stomatin can function as an anchor or organizer to initiate, or maintain, the formation of molecular complexes that internalize lipid ingredients from the extracellular environment (3). Stomatin may also modulate the function of effector molecules residing within the lipid rafts[52,53] by capturing or trapping the lateral diffusion of proteins and promote their interactions. Other fatty acid binding proteins (FABPs) may also participate in formation of this translocator complex and accelerate the internalization of LCFAs, resulting in an increased production of intracellular triglycerides (TG), that were then transported to, and stored in LDs (4). Stomatin may also involve in the latter process.

Stomatin has a unique hairpin-loop topology[54]. The hydrophobic intramembrane domain (25–54aa) near the N terminus contains a single conserved residue Pro-47, which determines stomatin's rare membrane monotopic structure[55]. Deletion of this N terminal hydrophobic pocket of stomatin slightly decreased FA uptake potency but did not affect LD growth. The C-terminal deletion mutant, on the other hand, can be oligomerized, be associated with lipid rafts[56–58], and is still capable of promoting LD growth and FA uptake (Supplementary Fig. 3b, c). Stomatin contains a conserved core region, the SPFH domain or stomatin-domain, that is shared by stomatin, prohibitin, flotillin, and HflK/

HflC protein families, the SPFH domain[59]. A small molecule OB-1 functions as an inhibitor for self-association of stomatin-domain proteins, including stomatin, STOML1, STOML2 and STOML3, but not podocin[38]. OB-1 treatments could effectively block LD growth and adipogenesis (Fig. 8).

Are above-mentioned stomatin's functions related to the protein's modulatory effects on cholesterol contents of the cell? As shown in Supplementary Fig. 2, stomatin-associated proteins include those involved in cholesterol transport, biosynthetic process and homeostasis; so, stomatin appears to be involved in cholesterol metabolisms. The filipin staining showed that free cholesterols were accumulated on LD surfaces and plasma membranes of adipocytes[60]. Increased expressions of stomatin or its truncated mutants did not affect the amounts and distributions of free cholesterol (Supplementary Fig. 4a), neither was the uptake of cholesterol influenced by excessive wild type or mutant stomatin proteins. On the other hand, stomatin knockdown significantly decreased the cholesterol content in adipocyte-like cells. The regulatory mechanisms for such inhibition are unclear. Decreased stomatin expressions also relocated CAV-1 from cell membranes to intracellular compartments (Supplementary Fig. 4b). CAV-1 is the main protein component of caveolae which are flask-shaped invaginations in the plasma membranes, and has been implicated in regulating cellular signal transduction, cholesterol homeostasis[61], and facilitating FA uptake by CD36-mediated caveolar endocytosis[36]. Whether stomatin's activities

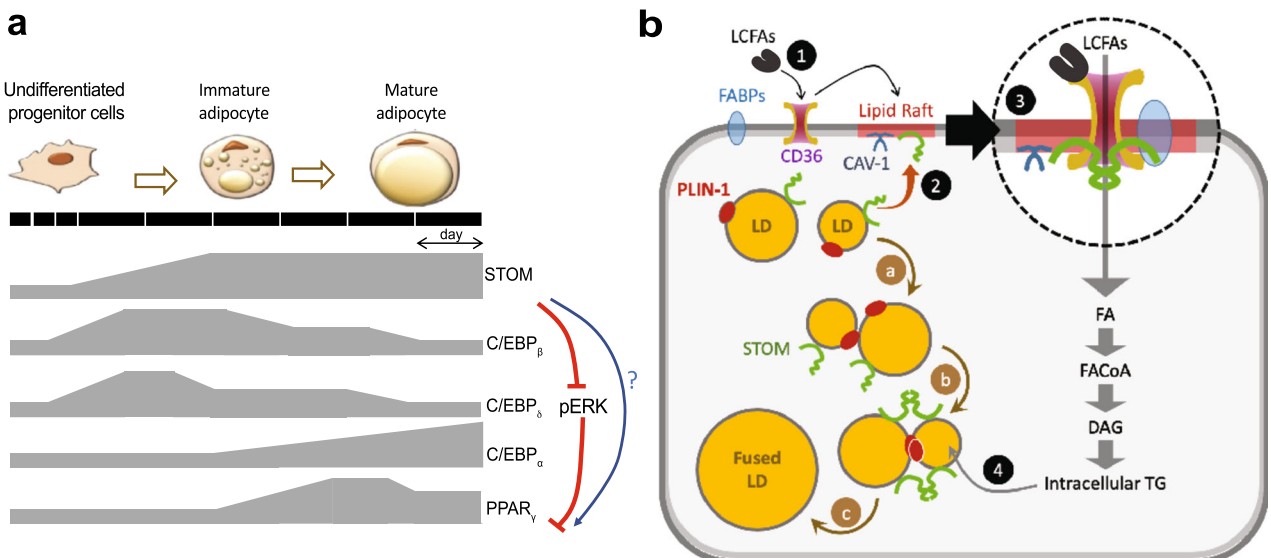

**Fig. 9 Schematic models illustrating dynamic gene expressions during induction of adipogenic differentiation and roles of stomatin in promoting fatty acid uptake and lipid droplet enlargement in adipocytes. a** Dynamics expression patterns of stomatin, adipogenic genes C/EBP$_\beta$, C/EBP$_\delta$, C/EBP$_\alpha$, and PPAR$_\gamma$ along the adipogenic differentiation process are depicted. Stomatin progressively increases during adipogenesis and may participate in modulation of other adipogenic gene. By inhibiting pERK, stomatin activates PPAR$_\gamma$, as a prelude to adipogenesis. Stomatin may also directly activate PPAR$_\gamma$ pathway through currently unknown mechanisms. **b** Stomatin and calveolin-1 are known to associated with lipid raft (shadowed red). Stomatin proteins are present on the surfaces of LDs, together with perilipin. Stomatin can promote enlargement of LDs through facilitating LD-LD fusion (a → b → c). The presence of LCFAs outside the cell, through currently unknown mechanisms, relocates stomatin proteins from LDs to lipid rafts on the plasma membrane (1→2), where they interact with CD36 and possibly other FABPs, to promote uptake of LCFAs (3), resulting in increase of intracellular triglycerides, which are transported and stored in LDs (4).

reported in this report related to the caveolae-mediated endocytosis is currently unknown.

In mice having excessive stomatin, the obese phenotype and functional impairments occur only when feeding the animal with high fat diet, instead of regular diet. These results shed light on how obesity is developed. With regular diet feeding, even the stomatin transgenic animal can keep its bodyweight in check because there is no superfluous energy supply. With HFD feeding, the excessive stomatin utilized surplus nutrients and energy to increase fat tissues. Fat mass expansion occurs by generating new fat cells (hyperplasia through adipogenesis) or hypertrophy of existing adipocytes, the relative balance of which may impact the systemic metabolism. It is noteworthy that superfluous circulating free fatty acid has harmful effects to the animal. Upregulation of stomatin, by promoting fatty acid uptake and LD growth, can help store excessive free fatty acid to the adipose tissue to protect the animals.

It is interesting to note that whole-body overexpression of human Stomatin did not promote adipogenesis in the visceral fat (VAT), but enhance more apparently adipocyte hypotrophy in the subcutaneous fat (SAT). Previous studies have shown that, during the initial phase of high fat diet (HFD)-induced obesity, VAT is the primary fat depot that expands. Such VAT expansion is followed by SAT to store the excess lipid[62]. Once the mice reached a bodyweight of around 40 g, the gonadal VAT expansion would be diminished[62]. This limited growth of VAT might be caused by higher turnover rates of VAT, compared to SAT. As a result, SAT hypertrophy was the major phenotype in adult mice with prolonged HFD feeding[63]. In addition, inappropriate hypertrophy of adipocytes reduces the tissue's regeneration capacity and increases metabolic complication of obesity[64,65]. Our results showed that surfeit stomatin expression promotes adipocyte hypertrophy in the subcutaneous fat, but not so much in visceral fat (Fig. 5a). This is supported by the finding that in SAT of individuals with obesity, increment of FAT/CD36 is

associated with a great capacity of raising the palmitate acid transport across the plasma membrane; such increase was not observed in VAT[66,67]. Further increases of FAT/CD36 expressions were noted in SAT of type2 diabetes patients[67].

As shown in Fig. 5a, since subcutaneous adipocyte hypotrophy was already very prominent in *STOM Tg* mice fed with HFD for a prolong period (i.e., 20 weeks), we could not rule out that other complications might have already occurred, such as direct effects of surplus stomatin on the liver and skeletal muscle, that might account for the functional abnormalities seen in HFD-fed *STOM Tg* animals. Indeed, stomatin transgenic mice also exhibited increased mass of lean tissues (Fig. 4f). This is due mainly to the fusogenic property of stomatin that causes not only adipocyte hypertrophy through LD-LD fusion, but also fusions of myoblasts[68,69], resulting in increased growth of skeletal muscles. More cell-type-specific studies are needed to clarify these issues.

In a broader sense, stomatin is involved in regulating various aspects of lipid homeostasis, including adipocyte differentiation, lipid production, lipid storage, lipolysis and lipid secretion. The finding that stomatin regulates fatty acid uptake and LD growth may provide new opportunities for correcting whole-body energy disorders or energy surplus-induced obesity by modulating the molecular events associated with stomatin.

## Methods

**Cell culture and induction of adipogenic differentiation.** 3T3-Ll murine fibroblasts, purchased from Bioresource Collection and Research Center (BCRC Number: 60159, Taiwan), were maintained in high-glucose Dulbecco's modified Eagle's medium (DMEM; Hyclone, SH30003.02) supplemented with 10 % bovine calf serum (BCS; Gibco, 16170-078) and 1% sodium pyruvate (Gibco, 11360-070),. To induce differentiation into adipocyte-like cells, 3T3-Ll cells were first grown to confluence. Two days after reaching confluence, the medium was replaced with DMEM supplemented with 10 % fetal bovine serum (FBS; Gibco, 26140-097) containing MDI cocktail, including 0.5 mM 3-Isobutyl-1-methylxanthine (IBMX; Sigma-Aldrich, SI-I7018), 0.25 μM dexamethasone (Sigma-Aldrich, D4902) and 10 g/ml bovine insulin. After 72 h, the medium was replaced with DMEM supplemented with 10 % FBS and 10 μg/ ml bovine insulin (Sigma-Aldrich, I6634).

The medium was refreshed every other day. Human embryonic kidney (HEK) 293 T cells (ATCC, CRL-3216) were cultured in DMEM supplemented with 10 % FBS.

**BSA-conjugated palmitic acid preparation and treatment**. Palmitic acid was conjugated to fatty acid–free (FFA) bovine serum albumin (Millipore and GoldBio) for fatty-acid induction experiments. Palmitic acid (PA; Sigma-Aldrich, P5585) was dissolved in 1 N NaOH to make a 2 mM PA stock solution. The stock solution of PA was incubated at 75 °C water bath for 2 h and then sterilized with a 0.22 μm filter. To prepare FFA-BSA solution, FFA-BSA was dissolved in serum-free DMEM and pre-warm to 37 °C, followed by sterilized with a 0.22 μm filter. PA-stock was then added to FFA-BSA solution and incubated at 55 °C water bath for 10 min[70]. The BSA-PA conjugated solution was diluted to 750 μM by adding 10% FBS DMEM for cell treatments.

**Lentivirus production and transduction**. Lenti-Vector pLAS2W.puro, pLAS4W. puro, shRNA for *Stom* vectors (shSTOM-1: clone number TRCN0000112911 and shSTOM-2: clone number TRCN 0000112912) and packaging plasmids pCMV delta R8.91 and PMD.G were obtained from National RNAi Core Facility Platform (Acadamic Sinica, Taiwan). GFP, hSTOM-GFP, RFP, hSTOM-RFP, hSTOM-Flag ΔC-hSTOM-Flag and ΔN-hSTOM-Flag were amplified from expression plasmids and cloned into pLAS2W.puro lenti-Vector. HEK-293T cells were co-transfected with Lenti-Vector and packaging plasmids using NTR II Non-liposome transfection reagent II (T-Pro-Biotechnology, Taiwan). The supernatants containing lentivirus were harvested after 24- and 48-h and centrifuged at 300 g for 5 min to exclude any remaining HEK-293T packaging cells. Subsequently, cells were transduced with the lentivirus-containing supernatants for 24 h, followed by 3 μg/mL puromycin selection for 3 days.

**Lipid droplets isolation**. Cells were washed and collected for lipid droplets isolation (Cell Biolabs, Inc. MET-5011). The cell pellets were suspended with 200 μl Reagent A. After 10-min incubation on ice, the mixture will be applied with 800 μl Reagent B for 10 min on ice. Homogenize the cells by passing them five times through a one-inch 27-gauge needle. Briefly spin the homogenate at 100 x *g* for 5 s and carefully layer 600 μL of 1X Reagent B on top of the homogenate, then spin for 3 h at 18000–20000 x *g* at 4 °C. The 270 μl (containing the floating lipid droplets) from the top of the tube were carefully collected and subjected to Western blotting and in vitro lipid droplet fusion assay.

**Quantifications of lipid accumulation**. Oil Red O stock solution was prepared by dissolving 0.35 g Oil Red O (no. O0625, Sigma Aldrich) in 100 ml pure isopropanol. Cells were fixed with 3.7% formaldehyde for 1 h at room temperature, followed by two-time wash, then dried entirely using a hairdryer. Post-fixed cells were stained with Oil Red O, diluted in distilled water (6:4) for 1 h at room temperature, further washed four-time with distilled water. The stained dyes in lipid droplets were extracted with pure isopropanol and absorbance at 490 nm (OD $_{490}$) was measured.

**Western blotting analysis**. Cultured cells were washed twice with cold PBS, then lysed using RIPA lysis buffer supplemented with cOmplete™ (EDTA-free Protease Inhibitor Cocktail) and PhosSTOP™ (Both obtained from Sigma Aldrich). The amount of total extracted proteins was quantified by Pierce 660 nm protein assay reagent (Thermo Fisher Scientific™). The protein lysates were separated by SDS-PAGE, then transferred onto PVDF membranes. The membranes were blocked with 5% bovine serum albumin (BSA) in TBS-T (Tris-buffer saline supplement with 0.1% Tween 20) at room temperature for 1 h, and then incubated with primary antibodies, such as anti-STOM (a166623b, Abcam), anti-perilipin (9349, Cell Signaling), anti- PPAR$_γ$ (2435, Cell Signaling), anti- C/EBP$_α$ (8178, Cell Signaling), anti-human STOM (E6, Santa Cruz Biotechnology), anti-Flag (CSB-MA000021M0m, Cusabio), anti-mCD36 (820263-T48, SinoBiological), anti-hCD36 (14347, Cell Signaling), anti-CAV-1 (7C8, Santa Cruz Biotechnology), anti-ATP1A1(3010, Cell Signaling) anti-β-Tubulin (T2200, Sigma Aldrich), and anti-Actin (GTX109639, GeneTex) antibodies, at 4 °C overnight. After washing three times with TBS-T, the PVDF membranes were incubated with horseradish peroxidase-conjugated secondary antibodies for 1 h at room temperature, then treated with substrates (SuperSignal West Femto Maximum Sensitivity Substrate, Thermo Fisher Scientific), then visualized by ImageQuant (GE Healthcare Life Science).

**Immunofluorescence**. Cells seeded on coverslips were fixed with 3.7% paraformaldehyde (Sigma-Aldrich) at room temperature for 15 min, rinsed twice with PBS for 5 min, permeabilized and blocked using extraction buffer composed of 0.1% saponin (Sigma-Aldrich) and 1 % BSA in PBS, for 1 h. Cells were then incubated with primary antibodies, including anti-STOM (a166623b, Abcam) and anti-perilipin (GT2781, GeneTex), anti-CAV-1 (N20, Santa Cruz Biotechnology) at 4 °C overnight or incubated with AF647-mCD36 (MF3, Bio-Rad) at RT for 2 h. After washing three times with wash buffer (PBS with 0.1% saponin), cells were

incubated with fluorochrome-conjugated secondary antibodies for 1 h, and observed under a confocal microscope (LSM700, Zeiss).

**Immunoprecipitation**. 3T3-L1 cells over-expressing hSTOM-Flag were induced to differentiate and the cells were harvested and collected the supernatant of lysate after at least 2 h lysis on ice (Tris Hcl pH 7.5 25 mM; NaCl 150 mM; EDTA 1 mM; Triton 0.1%; Na-deoxycholate 0.5%). 293 T cells were co-transfected with constructs of hSTOM-Flag and human CD36. After 72 h of transfection, the cells were harvested and collected the supernatant of lysate after at least 2 h lysis on ice. The lysate supernatant was collected after centrifuged at 16,100 g for 20 min at 4 °C. Clear protein lysates were incubated with flag antibody-conjugated agarose beads overnight at 4 °C. Agarose beads were washed three times with lysis buffer and centrifuge at 6000 g for 1 min. The bound proteins were eluted by incubating at 37 °C in 2X sampling dye for 10 min. Samples were obtained after centrifuging at 800 g for 2 min and boiled for 5 min at 95 °C. Eluted lysate was subjected to SDS-PAGE and detected by LC-MASS or immunoblotting with the antibodies.

**Protein digestion and peptides extraction**. After immunoprecipitation, the eluted proteins were separated by gradient SDS-gel. After being stained with R250 for 4 h, the gel was de-stained by the solution of 50% ACN (50% of ACN in ddH$_2$O) and cut into six segments. Each of the segments was conducted with in-gel digestion. First, the gel was dried in a Speed Vac (Thermo, SAVANT, RVT5105). After that, the gel was incubated in beta-mercaptoethanol and 4-vinyl pyridine for 20 min at dark, respectively. Gel pieces were subsequently washed twice with 25 mM ammonium bicarbonate (pH 8.5) for 15 min. Next, the gel pieces were vacuum dried and digested with 25 ng sequencing grade trypsin (Promega, Madison, WI) in 25 mM ammonium bicarbonate (pH 8.5) at 37 °C for overnight. After digestion, the tryptic peptides were extracted twice with 50% acetonitrile containing 0.1% formic acid for 15 min with sonication and then dried under a vacuum.

**Liquid chromatography tandem MS (LC-MS/MS) protein identification**. The dried pellet was dissolved in 10 ml of 0.1% formic acid and then subjected to an LTQ-VELOS mass spectrometer with a nano-electrospray ionization source (Thermo Electron, MA, USA) coupled to a nanoAcquity ultra Performance LC (UPLC) system (Waters, Manchester, UK). Protein peptides were loaded into UPLC, then were captured and desalted on a C18 trap column (180 × 20 mm; Waters, USA). After that, protein peptides were further separated by a BEH C18 column (25 cm × 75 μm, Waters, USA). The mobile phases were prepared as solution A (0.1% formic acid in ddH$_2$O) and solution B (0.1% formic acid solution in acetonitrile). The separation condition was performed by eluting the peptides from the column with a linear gradient of 5–40% solution B for 90 min, 40–95% solution B for 5 min, 95% solution B for 10 min at a flow rate of 0.3 μL/min. The eluted protein peptides were ionized and delivered into LTQ-VELOS mass spectrometer. The mass spectra data of peptides were acquired using a data-dependent acquisition method. One full MS survey scan (m/z: 200–1500) at a resolution of 30,000 was followed by an MS/MS scan of the ten most intense multiply charged ions (2+ and 3 + ). The MS/MS data were then analyzed by PEAKS Studio 8.5 (Bioinformatic Solution, Ontario, CA). The search was conducted against the UniProt mouse protein database (containing 17,089 protein sequences; released on July 2021; http://www.uniprot.org/). A protein was identified as at least one unique peptide was matched—quantitative analysis of proteins by MS spectra counting with in-house software[71]. Source data are provided with this paper.

**FRAP assay**. Adipocyte-like 3T3-L1 cells were incubated with 6 μM of Bodipy™-FL-C$_{12}$ (D3822, Invitrogen™) at 37 °C overnight. After refreshing with complete medium, live cells were viewed under a confocal microscope (Zeiss, LSM700) using a 100x oil immersion objective. Selected regions were first bleached with 15 pulses of 100% laser power (combined 488 with 405 diode lasers), followed by time-lapse recording at 30-s interval using normal imaging laser power.

**Cell surface biotinylation**. Cell were incubated with fresh prepared sulfo-NHS-SS-biotin (0.5 mg/ml) for 30 min at 4 °C. After wash cell with chill-PBS containing 100 mM glycine, cells were lysed by PBS supplement with 1% Tritonx-100 for 1 h on ice. The clear lysates were collected and incubated with streptavidin agarose at 4 °C overnight. Beads were washed and eluted with 2x SDS loading buffer at 37 °C for 1 h, and subjected to Western blot assay.

**Antibodies**. Perilipin (#9349), PPAR gamma (#2435), C/EBP alpha (#8178), pERK (#4370), ERK (#4965), pAkt (#4060), Akt (#4691), lipolysis Activation Antibody sampler kit (#8334), hCD36 (#14347) and ATP1A1(#3010) from Cell Signaling. STOM(a166623b) from Abcam. Beta-Tubulin(T5201) from Sigma Aldrich. Flag (CSB-MA000021M0m) from Cusabio. mCD36 (820263-T48) from SinoBiological. beta-Actin (GTX109639), Perilipin (GTX2781) from GeneTex. Human STOM (M14), Human STOM(E6), CAV-1 (7C8), CAV-1 (N20), CD36 (SM), GFP (FL) from Santa Cruz Biotechnology. AF647-632 mCD36 (MF3) from Bio-Rad. All primary antibodies were used at a dilution of 1:1000 for immunoblotting and 1:100~200 for immunofluorescent staining. Alexa Fluor® 647 AffiniPure Goat

Anti-Rabbit IgG (H + L)(#111-605-003), Rhodamine Red™-X (RRX) AffiniPure Goat Anti-Mouse IgG (H + L)(#115-545-003), Rhodamine (TRITC) AffiniPure Goat Anti-Rabbit IgG (H + L)(#111-025-144) and Peroxidase conjugated Goat anti-Mouse IgG (H + L) (#115-035-003), Peroxidase conjugated Goat anti-Rat IgG Fc Fragment specific (#112-035-071) from jackson ImmunoResearch. Peroxidase conjugated Goat anti-Rabbit IgG (H + L) (Ap132P) from Chemicon®. Peroxidase conjugated Donkey anti-Goat IgG (H + L) (sc-2020) from Santa Cruz Biotechnology. VeriBlot for IP Detection Reagent (HRP) (ab131366) from Abcam. All secondary antibodies were used at a dilution of 1:5000~10000 for immunoblotting and 1:200~400 for immunofluorescent staining.

**Proximity ligation assays (PLA).** Cells were fixed by 4% paraformaldehyde 15 min, followed by permeabilization and blocking procedures. Cells were incubated with primary antibodies (1:100 dilution), CD36 (SMφ, Santa Cruz Biotechnology) x GFP (FL, Santa Cruz Biotechnology) and CAV-1 (N20, Santa Cruz Biotechnology) x hSTOM (E6, Santa Cruz Biotechnology) at 4 °C overnight. After washing cells by 1x Wash Buffer A, the cells were incubated with the PLUS and MINUS PLA probes for 1 h at 37 °C. Tap off the PLA probes solution and wash cells with Wash Buffer A twice. The ligation solution was applied and incubated with cells for 30 min at 37 °C. After the wash step, cells were incubated with the amplification solution for 100 min at 37 °C. Finally, wash cells two-time using 1x Wash Buffer B, followed by one-time 0.01x Wash Buffer B. The cells were mounted and analyzed in a confocal microscope.

**Fatty acid uptake assay in vitro and in vivo.** For in vitro analysis, adipocyte-like 3T3-L1 cells were serum-starved in serum-free DMEM containing 1% sodium pyruvate for 1 h at 37 °C/5% CO2, changed to serum-free DMEM containing 1% sodium pyruvate and 10 μg/ml insulin for 30 min, then to 1X HBSS containing 20 mM HEPES. Subsequently, 0.2 μM Bodipy-FL-$C_{16}$ (Thermo) was added. The fluorescence (excitation: 485 nm and emission: 515 nm) was detected by ELISA reader (TECAN) using bottom-read mode and kinetic reading at a 30 s interval. For in vivo analysis, All the animal's experiments were approved by Institutional Animal Care and Use Committee (IACUC) of National Yang Ming Chio Tung University. (IACUC #1050801, 1050801r,1090706,1090706r). 400 nM Bodipy-FL-$C_{16}$ was emulsified in 65 mg/ml BSA solution to prepare the probe solution. An amount of 150 μl probe solution was injected into the animal via tail vein. The mice were sacrificed and their subcutaneous adipose tissues were resected for quantifications of fluorescence. Fat pads were homogenized using homogenizer (Roche MagNA Lyser Benchtop Homogenizer) for 45 s at 3000 r.p.m., and incubated at 65 °C for 30 min. After centrifugation at 13,000 g for 10 min, fat layer was collected and subjected to fluorescence signal quantifications.

**STOM transgenic and Stom knockout mice generation.** All the animal's experiments were approved by Institutional Animal Care and Use Committee (IACUC) of National Yang Ming Chio Tung University. (IACUC #1050801, 1050801r,1090706,1090706r). Transgenic mice overexpressing human STOM were generated by pronuclear microinjection. STOM was cloned into vector (STOM-p1033), and RNA polymerase II large subunit promoter was used to drive expression in C57BL/6 mice. The integration of the transgene was confirmed by PCR analysis of mouse tail DNA.

Stom knock out mice were generated by cre-loxp strategy[72]. Mice lacking exon 2–4 of the Stom locus were generated using a targeting strategy that deleted the genomic sequence containing the murine Stom exon 2–4, resulting in out-of-frame splicing between exons 1 and 5 (Supplementary Fig. 8a). Recombinant animals identified by PCR genotyping, were crossed with C57BL/6-expressing Cre-recombinase-expressing mice (ZP3-Cre), and offspring were genotyped by PCR to identify those that were heterozygous $Stom^{+/-}$ for the knockout genotype. Heterozygous knockout animals were backcrossed to C57BL/6 mice to generate congenic $Stom^{-/-}$ mice.

All of the mice were housed on a 12 h light/dark cycle at 22 °C.

**Mice phenotype analyses.** All the animal's experiments were approved by Institutional Animal Care and Use Committee (IACUC) of National Yang Ming Chio Tung University. (IACUC #1050801, 1050801r,1090706,1090706r). Three to four weeks male genetically engineered mice and wild type littermates were fed with either chow diet (Laboratory Autoclavable Rodent Diet 5010*) or high-fat diet (D12492, 60 kcal% fat, Research Diets) for up to 20 weeks under free-feeding conditions. All of the mice were housed on a 12-h light/dark cycle at 22 °C. The blood biochemistry of mice were examined: fasting blood glucose and plasma insulin was determined by human blood glucose meter (Accu-chek Performa, Roche) and Mouse Ultrasensitive Insulin ELISA (80-INSMSU-E01, ALPCO), respectively. GOT and GPT were measured using Automated Clinical Chemistry Analyzer (FUJI DRI-CHEM 4000i).

**Glucose tolerance tests.** All the animal's experiments were approved by Institutional Animal Care and Use Committee (IACUC) of National Yang Ming Chio Tung University. (IACUC #1050801, 1050801r,1090706,1090706r). Mice fasted overnight were injected intraperitoneally with glucose solution (1 g/kg body weight). Blood glucose levels were measured at 0, 15, 30, 60, and 120 min after injection using a glucometer.

**Histological analyses.** The liver and adipose tissues were formalin-fixed and paraffin-embedded. Sections of WAT and the liver were stained with hematoxylin and eosin (H&E stain). All tissue images were obtained using the Aperio CS2 Digital Pathology Scanner (Leica) and analyzed via Imagescope software. The size of adipose cells was analyzed using the Fiji Adiposoft software to obtain the area of each and individual adipocyte.

**Transcriptome expression analyses.** Total RNA was extracted from cells using TRIzol® reagent and subjected to global transcriptome analysis using Mouse Genome Arrays (MTA-1_0, Affymetrix). The Transcriptome Analysis Console (TAC4.0) was used to process and analyze CEL files. We initially filtered for 11,264 probe sets annotated as locus type "coding". Subsequently, 1478 annotated coding genes were identified via exclusively a stringent threshold of $p < 0.05$ and FDR $p < 0.001$. Additionally, we further filtered for 379 genes via exclusively a stringent fold change threshold ≥3, or ≤ −3.

Source data are provided with this paper. The results were confirmed using qRT-PCR. The cDNAs were generated using SuperScript III First-Strand Synthesis System following the manufacturer's protocol. To determine the expression of specific genes, quantitative real-time PCR (qRT-PCR) was performed using diluted cDNA in a total volume of 5ul with SYBR Green (Qiagen). Gene expression was normalized to the internal reference gene Nono, followed by calculation using Δ ΔCT method. Sequences of primers are provided as a separated Supplementary Table 1.

**Direct adipose injection of AAV.** AAV U6 expression vector is inserted with a promoter for human U6 to drive Stom gene knockdown, which carried the reporter GFP driven by the PGK promoter. Serotype 8 AAV(AAV8) vectors encoding mouse Stom shRNAs were produced via the AAV Core of IBMS, Academia Sinica. AAV8 with CMV promoter-driven expression of GFP as control were harvested from the RNAi core. After anesthetizing mice with isoflurane, we cleaned the skin of mice with ethanol and made a 0.5 cm incision using surgical scissors. Hold skin open with tweezers to expose the fat pad. Take the injection device, carefully insert the needle into the fat pad, and start injecting 30 μl AAV8 ($2.0 \times 10^{11}$ VG) into multiple distinct spots in the fat pad using the fine needle. After the virus solution is injected completely, carefully take out the needle and dispose of the needle together with the tubing according to local regulations. Close the incision with Michel's suture clips.

**Statistics and reproducibility.** Statistical analysis was performed using GraphPad Prism 7. Data with error bar are presented as mean value ± s.d or s.e.m. One-way, Two-way ANOVA, Paired or unpaired Student's two-sided t test and multiple unpaired t test (A = 0.05, Definition of statistically significant) was used to determine the p value. Differences were considered statistically significant when the p value was <0.05. Results in Fig. 1b-e; 2b, d; 3a, e, f; 4b, c; 6a, b; 7a; 8d; Supplementary Figs. 3a; 4e; 5a; 8f. are representative data of three independent repeats. And there were similar results in three independent repeats.

**Reporting summary.** Further information on research design is available in the Nature Research Reporting Summary linked to this article.

## Data availability
The expression profiling microarray data has been deposited in public Gene Expression Omnibus (GEO) database under the accession codes GSE205011. The mass spectrometry proteomics data have been deposited to the ProteomeXchange Consortium via the PRIDE[73] partner repository with the dataset identifier PXD031866 and protein identification is provided in Supplementary Data 1. Source data are provided with this paper.

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

## Acknowledgements

The authors acknowledge Mouse Genome Arrays analysis provided by Genomics Center for Clinical and Biotechnological Applications (GCCBA) in Cancer Progression Research Center, NYCU. GCCBA is a National Core Facility for Biopharmaceuticals (NCFB) that is funded by Ministry of Science and Technology, Taiwan. We thank the technical services for protein identification from Proteomic Research Center and Cancer Progression Research Center (NYCU). We thank the technical services provided by the "Transgenic Mouse Model Core Facility of the National Core Facility Program for Biotechnology, National Science Council" and the "Gene Knockout Mouse Core Laboratory of National Taiwan University Center of Genomic Medicine". We thank Dr. Li-Ru You (NYCU) provided the materials for generating *Stom* null mice. We thank Taiwan Mouse Clinic (TMC) of Academia Sinica and Taiwan Animal Consortium for technical support and Small Animal Imaging Core Facility for in vivo imaging. We thank to National Center for High-performance Computing (NCHC) for providing computational and storage resources. We are deeply grateful to Dr. Chian-Feng Chen (NYCU), Tzuu-Shuh Jou (NTU), Jia-Fwu Shyu (NDMC), Hong-Yu Chien (TCH) and Ke-Hsun Hsu (NYCU), for their scientific suggestions; Dr. Li-Li Li (LTRI, Canada) for her helpful proof-reading of the manuscript. This work was financially supported by the "Cancer Progression Research Center, National Yang Ming Chiao Tung University" from The Featured Areas Research Center Program within the framework of the Higher Education Sprout Project by the Ministry of Education (MOE) in Taiwan. This work was supported by the Ministry of Science and Technology: MOST 110-2634-F-A49-005 and MOST 110-2740-B-A49A-501. The authors declare no conflicts of interest. NYCU: National Yang Ming Chiao Tung University. NTU: National Taiwan University. NDMC: National Defense Medical Center. TCH: Taipei City Hospital.

## Author contributions

S.-C.W., W.-N.L., C.-Y.T., and C.-H.L. designed the research; S.-C.W., C.-C.L., and Y.-M.L. performed the research and analyzed the data; S.-C.W., J.-H.L., H.-W.L., and T.-W.C. contributed microscopic approaches; W.-J.L., C.-Y.C., K.-H., and C.-Y.Y. contributed materials; and S.-C.W., and C.-H.L. wrote the manuscript.

## Competing interests

The authors declare no competing interests.
