## [Peer Review File · Nature Communications]

Stomatin modulates adipogenesis through the ERK pathway and regulates fatty acid uptake and lipid droplet growthReviewers' Comments:

Reviewer #1:

Remarks to the Author:

This manuscript focused on the role of Stomatin on in vitro adipogenesis and lipid droplet growth, as well as high fat diet induced fat expansion. The authors found that Stomatin expression is increased during in vitro adipogenesis of 3T3-L1 cells. Over-expression of human Stomatin increased the lipid droplet size in 3T3-L1 adipocytes, and knockdown of Stomatin inhibited adipogenesis and lipid droplet formation. In the in vivo model, whole-body human Stomatin overexpression led to enhanced body fat gain during high fat diet feeding. These mice had increased adipocyte cell size in subcutaneous fat, fatty liver, and whole-body insulin resistance. The author also showed that in vitro, knockdown of Stomatin activated the ERK pathway. They concluded that Stomatin positively regulates both adipogenesis and lipogenesis in adipocytes.

The topic of this study is of great interest to the field of adipose biology. The authors' discovery is novel and important. However, the major conclusion of the manuscript comes from in vitro study in 3T3-L1 cells, and the data presented in the current version do not fully support the authors' conclusions. Here are some specific comments that will hopefully help to improve the manuscript.

1. In the in vivo diet induced obesity model, we usually see adipogenesis in the visceral fat and adipocyte hypotrophy in the subcutaneous fat. One of the major concerns is that whole-body overexpression of human Stomatin did not promote adipogenesis in the visceral fat but only promoted adipocyte hypotrophy in the subcutaneous fat. Please add more discussions about how these two depots are different. Please also discuss adipogenesis/lipid droplet formation and lipid droplet enlargement in mature adipocytes separately, based on these results.
2. It would be important to show how Stomatin would alter its expression during high fat diet feeding in each fat depot.
3. Will Stomatin knockdown in the subcutaneous fat (e.g., through local virus injection) prevent high fat diet induced adipocyte hypotrophy?
4. Are the metabolic disorders caused majorly by subcutaneous adipocyte hypotrophy? Whole-body Stomatin over-expression is complicated and should have many direct effects on the liver and skeletal muscle. The lean mass in these mice is also increased. The phenotype of these mice is robust and interesting. Some cell-type-specific study is highly recommended for future studies.
4. In Figure 1, there are huge areas of mSTOM positive signals that are not associated with lipid droplets or perilipin, despite the authors' descriptions.
5. In the introduction, please cite more original papers. For example, for high fat diet induced hypertrophy vs. hyperplasia; for C/EBP α and PPAR γ as master regulators for adipogenesis.

Reviewer #2:

Remarks to the Author:

Stomatin is a well-known lipid raft protein and may bind cholesterol directly through the conserved SPFH domain. It also has a hydrophobic hairpin that suits LD surface monolayer well. In this manuscript, the authors investigated the effects of stomatin overexpression or knockdown on adipogenesis and LD dynamics. Overexpressing stomatin increases adipogenesis, fatty acid uptake, and LD growth in L1 cells. Overexpressing stomatin in mice promoted obesity, insulin resistance and steatosis. Knocking down stomatin or using inhibitors in L1 cells had the opposite effects to overexpression. The effects of stomatin may involve ERK signaling, although other mechanisms also exist. Overall, the experiments were carried out carefully and the data were presented very well. Good phenotypic characterization. A major weakness is a lack of mechanistic explanation. Also, these effects on adipogenesis and LD dynamics are fairly common, i.e. manipulating hundreds if not thousands of genes would lead to similar mild effects on adipose tissue as reported here. Also some controls would help.

Specific concerns are listed below.

1. In all overexpression experiments, a nice control would be a known lost-of-function mutant. Is

there such a stomatin mutant that abolishes its interaction with cholesterol? What about mutating the proline that is important to the hairpin? Such a mutant should be used in cell experiments.

2. Figure 1C, immunoEM or Apex-EM is required to better understand stomatin and LDs.

3. Figure 5A. A typical differentiation assay should include qRT-PCR of key adipogenic genes such as *pref-1*, *ap2*, *adipsin* etc.

4. Mechanism of action. This is a major weakness. Given stomatin is a raft protein, the authors should examine caveolae in adipocytes in KD cells. Both caveolin-1 and cavin 1 are linked to lipodystrophy, and caveolin-1 is a LD protein. Also, CD36 level and localization need to be examined. CD36 localizes to rafts and interact with LDs (see PMID: 32958780). It is likely that stomatin regulate adipogenesis and LD dynamics through caveolae or CD36.

5. What happens to cholesterol in these O/E and KD cells? Cholesterol in adipocytes can impact adipogenesis and LD dynamics (see PMID: 31727739). Given its role in raft, stomatin may regulate adipogenesis and LD dynamics through cholesterol.

Minor points:

The first sentence of the abstract is grammatically incorrect. "is" not "are"

The authors need to be careful with the term lipogenesis, which strictly speaking, is the conversion of glucose to fatty acids. Should use TAG synthesis instead.

Line 138, membranes of LDs? What does it mean? LDs do not have a membrane as they contain only a monolayer.

Ref. 30 is wrong. No one can repeat that data.

Reviewer #3:

Remarks to the Author:

This is a highly interesting study on the function of stomatin, an enigmatic, lipid raft-associated, monotopic, integral membrane protein with peculiar structure that is highly conserved during evolution from archaea to humans. The authors found that stomatin is an important player in adipocyte differentiation and lipogenesis; it is involved in fatty acid uptake and growth of lipid droplets (LDs).

Transgenic stomatin overexpressing mice, in contrast to control mice, when fed with high fat diet, were obese and insulin-resistant; they showed hepatomegaly, hepatic dysfunction and steatosis. From these data one can clearly infer that stomatin is a major component of adipogenesis.

•What are the major claims of the paper?

(1)The "lipid raft" protein stomatin participates in adipogenesis and lipogenesis by preferentially recruiting effectors to lipid droplets (LDs) to induce fatty acid uptake and LD fusion. Adipocytes with increased stomatin expression exhibit higher levels of fatty acid uptake and LD growth or enlargement.

(2)Transgenic mice expressing human stomatin that were fed with a high-fat diet showed increased stomatin expression that facilitated progression of obesity, caused insulin resistance and hepatic impairments.

(3)Inhibition of stomatin by gene knockdown or pharmacological treatment blocked not only LD growth but also adipogenic differentiation by downregulation of the PPAR γ pathway.

(4)Effects of stomatin on PPAR γ involved ERK signaling; however, an alternate pathway also exists.

(5)Amongst various anti-obesity measures, stomatin serves as another potential therapeutic target.

•Are the claims novel?

The claims are absolutely novel. Stomatin has been identified on LDs before but the functional role of stomatin in adipocytes or on LDs has not been shown previously. Nothing was known about the involvement of stomatin in the PPAR γ pathway. Nothing related has been reported about stomatin knockout mice.

•Will the paper be of interest to others in the field?

Certainly, this paper will be of great interest to researchers in many fields, such as biochemists, pharmacologists, nutrition scientists and general physicians. The global obesity epidemic is one of the major concerns of society and therefore I believe that the paper will be highly interesting also for a wider public and probably also for pharmaceutical companies.

•Will the paper influence thinking in the field?

The paper will have a great impact on the thinking in the field of adipogenesis and the mechanism of LD generation. Stomatin has not been a player in this context before. The paper will also stimulate research in the caveolin field, because caveolins, which are structurally similar, have been identified on LDs but their function on LDs is still unknown.

•Are the claims convincing? If not, what further evidence is needed?

In general, the claims appear convincing. The data appear sound. In detail, the data showing LD-vesicle fusion could be more convincing, particularly the notion that small LDs are fusing with large LDs appears more incidental. It is not known, whether these small vesicles are really LDs.

•Are there other experiments that would strengthen the paper further? How much would they improve it, and how difficult are they likely to be?

The existence of tiny LDs could be easily verified by immunofluorescence co-staining with anti-stomatin and anti-perilipin (or another LD marker) and showing co-localization.

•Are the claims appropriately discussed in the context of previous literature?

In my view, the claims are appropriately discussed in the context of literature. The claim that stomatin serves as a potential therapeutic target, appears somewhat premature.

•Is the manuscript clearly written? If not, how could it be made more accessible?

The manuscript is clear, however, it should be undergone further careful proofreading regarding typos and grammar. On p.9, line 245, the headline says "Somatin" instead of "Stomatin".

•Could the manuscript be shortened to aid communication of the most important findings?

The manuscript is packed with data derived from studies using various methods. I do not see a possibility to shorten any of these studies.

•Have the authors done themselves justice without overselling their claims?

In my view, the claims made are justified.

•Have they been fair in their treatment of previous literature?

As far as my field is concerned, they had been fair in citing the relevant literature.

•Have they provided sufficient methodological detail that the experiments could be reproduced?

The methods are described in detail and should be reproducible.

•Is the statistical analysis of the data sound?

The statistical analyses appear sound.

•Should the authors be asked to provide further data or methodological information to help others replicate their work?

The methods are more or less standard methods. I do not see the necessity for more detailed information.

•Are there any special ethical concerns arising from the use of animals or human subjects?

Certainly, many mice have been sacrificed during this study, however, at a certain point there is no alternative to animal experiments, particularly in biomedical research.

Rainer Prohaska

Responses to the Reviewer's Comments

Reviewer #1:

This manuscript focused on the role of Stomatin on in vitro adipogenesis and lipid droplet growth, as well as high fat diet induced fat expansion. The authors found that Stomatin expression is increased during in vitro adipogenesis of 3T3-L1 cells. Over-expression of human Stomatin increased the lipid droplet size in 3T3-L1 adipocytes, and knockdown of Stomatin inhibited adipogenesis and lipid droplet formation. In the in vivo model, whole-body human Stomatin overexpression led to enhanced body fat gain during high fat diet feeding. These mice had increased adipocyte cell size in subcutaneous fat, fatty liver, and whole-body insulin resistance. The author also showed that in vitro, knockdown of Stomatin activated the ERK pathway. They concluded that Stomatin positively regulates both adipogenesis and lipogenesis in adipocytes. The topic of this study is of great interest to the field of adipose biology. The authors' discovery is novel and important. However, the major conclusion of the manuscript comes from in vitro study in 3T3-L1 cells, and the data presented in the current version do not fully support the authors' conclusions. Here are some specific comments that will hopefully help to improve the manuscript.

Comment 1-1. In the in vivo diet induced obesity model, we usually see adipogenesis in the visceral fat and adipocyte hypotrophy in the subcutaneous fat. One of the major concerns is that whole-body overexpression of human Stomatin did not promote adipogenesis in the visceral fat but only promoted adipocyte hypotrophy in the subcutaneous fat. Please add more discussions about how these two depots are different. Please also discuss adipogenesis/lipid droplet formation and lipid droplet enlargement in mature adipocytes separately, based on these results.

Response:

Thank you very much for the suggestions.

We have added a paragraph in Discussion section on p17-18 (**red texts**):

“It is interesting to note that whole-body overexpression of human Stomatin did not promote adipogenesis in the visceral fat (VAT), but enhance more apparently adipocyte hypertrophy in the subcutaneous fat (SAT). Previous studies have shown that, during the initial phase of high fat diet (HFD)-induced obesity, VAT is the primary fat depot that expands. Such VAT expansion is followed by SAT to store the excess lipid⁶². Once the mice reached a bodyweight of around 40 g, the gonadal VAT expansion would be diminished⁶². This limited growth of VAT might be caused by higher turnover rates of VAT, compared to SAT. As a result, SAT hypertrophy was the major phenotype in adult mice with prolonged HFD feeding⁶³.”

See also in the first paragraph on p18 (*lines 523-527*):

“..... in SAT of obese individuals, increment of FAT/CD36 is associated with a great capacity of raising the palmitate transport across the plasma membrane; such increase was not observed in VAT^{66, 67}. Further increases of FAT/CD36 expressions were noted in SAT of type2 diabetes patients⁶⁷.”

In addition, we have reorganized Results section, such that adipogenesis/lipid droplet formation and lipid droplet enlargement are separately described in the manuscript and summarized in the two working hypotheses depicted in Fig. 9A and 9B.

Comment 1-2. It would be important to show how Stomatin would alter its expression during high fat diet feeding in each fat depot.

Response:

In response to the reviewer’s comment, new experiments were performed and the results are shown in Fig. 4A.

The mRNA of mSTOM increased in both subcutaneous (SAT) and visceral (VAT) white adipose tissues in HFD-fed mice, compared to CD-fed mice, but such increase was less apparent in brown adipose tissue (BAT).

Comment 1-3. Will Stomatin knockdown in the subcutaneous fat (e.g., through local virus injection) prevent high fat diet induced adipocyte hypertrophy?

Response:

In response to the comment, new experiments were performed and the results are shown in Supplemental Fig. 6, and described on p11 (**red texts**).

“To knockdown stomatin locally at fat tissues instead of whole body, we injected AAV virus carrying shSTOM/GFP or GFP at SAT fat pad of mice (Supplemental Fig. 6A). The targeted SAT tissues were dissected and analyzed for their masses and histological sections (Supplemental Fig. 6B). We found that after HFD-feeding, the mass of AAV-infected SAT was slightly lower than the control SAT (Supplemental Fig. 6C). Using immunostaining of Na-K-ATPase for cell size measurements, we also noticed that the cells at the local shSTOM/GFP injected tissue were smaller than the cells on the tissues injected with control GFP vector (Supplemental Fig. 6D).”

Comment 1-4. Are the metabolic disorders caused majorly by subcutaneous adipocyte hypotrophy? Whole-body Stomatin over-expression is complicated and should have many direct effects on the liver and skeletal muscle. The lean mass in these mice is also increased. The phenotype of these mice is robust and interesting. Some cell-type-specific study is highly recommended for future studies.

Response:

Thank you for the comment. Indeed, phenotypes associated with whole body overexpression need to be interpreted with care. This consideration is added, in response to the critique, in the Discussion section (**red texts** of second paragraph, p18):

“As shown in Fig. 5 A, since subcutaneous adipocyte hypotrophy was already very prominent in *STOM Tg* mice fed with HFD for a prolonged period (i.e., 20 weeks), we could not rule out that other complications might have already occurred, such as direct effects of surplus stomatin on the liver and skeletal muscle, that might account for the functional abnormalities seen in HFD-fed *STOM Tg* animals. Indeed, stomatin transgenic mice also exhibited increased mass of lean tissues (Fig. 4F). This is due mainly to the fusogenic property of stomatin that causes not only adipocyte hypertrophy through LD-LD fusion, but

also fusions of myoblasts^{68,69}, resulting in increased growth of skeletal muscles. More cell-type-specific studies are needed to clarify these issues.”

Comment 1-5. In Figure 1, there are huge areas of mSTOM positive signals that are not associated with lipid droplets or perilipin, despite the authors' descriptions.

Response:

Thank you very much for the correction. The results of Fig. 1 have been rewritten as in the following (red texts of third paragraph, p5):

“Immunofluorescence staining revealed subcellular distributions of stomatin. In addition to puncta staining in the cytosol, stomatin proteins were noted on the plasma membranes (*arrows*), as well as surfaces of LDs (*inset*, Fig. 1C), where they partially colocalized with perilipin proteins when examined under STED microscopy (Fig. 1D). When LDs were isolated from adipocyte-like cells, stomatin was present in the LD fraction, together with the known LD-associated protein perilipin (Fig. 1E).”

Comment 1-6. In the introduction, please cite more original papers. For example, for high fat diet induced hypertrophy vs. hyperplasia; for C/EBPa and PPARg as master regulators for adipogenesis.

Response:

Thank you very much for the suggestions. New texts are added in the Introduction section (red texts, p4-p5)

“The expansion of adipose depots, especially the white adipose tissues, are characterized by the increase in adipocyte size (hypertrophy), or by formation of new adipocytes from the precursor cells (hyperplasia)^{26, 27, 28}. In the presence of excessive energy, mature adipocytes increase in cell size and undergo cellular hypertrophy to store the surplus fat²⁹. Hypertrophic adipocytes are characterized by excessive growth of LDs; the resulting unilocular LD may occupy more than 90% of the cell volume³⁰. The hypertrophic adipocytes are responsible for dysfunction of lipid homeostasis, along with pathological consequences³¹; while adipocyte hyperplasia plays a role in preventing

hypertrophy development and further maintaining the normal function of adipose tissue³². Approaches aimed at increasing adipogenesis or adipogenic differentiation, over adipocyte hypertrophy are now regarded as means to treat metabolic diseases. Notably, adipocyte expansion through adipogenesis could mitigate the negative metabolic effects of obesity³³.”

More references for regulations of adipogenesis are included in the Discussion section (red texts of second paragraph, p14):

“As shown in Fig. 9A, an undifferentiated progenitor cell can be induced to differentiate into an immature adipocyte, then to a mature adipocyte. Our results demonstrate a transient increase of two early adipogenic differentiation genes³⁹, C/EBP_β and C/EBP_δ during adipogenic differentiation. The rise of C/EBP_α, however, occurs at a relatively later phase³⁹. These adipogenic genes seem to converge to PPAR_γ, which serves as a master regulator for signaling pathways that lead to adipocytic differentiation⁴⁰.”

Reviewer #2 (Remarks to the Author):

Stomatin is a well-known lipid raft protein and may bind cholesterol directly through the conserved SPFH domain. It also has a hydrophobic hairpin that suits LD surface monolayer well. In this manuscript, the authors investigated the effects of stomatin overexpression or knockdown on adipogenesis and LD dynamics. Overexpressing stomatin increases adipogenesis, fatty acid uptake, and LD growth in L1 cells. Overexpressing stomatin in mice promoted obesity, insulin resistance and steatosis. Knocking down stomatin or using inhibitors in L1 cells had the opposite effects to overexpression. The effects of stomatin may involve ERK signaling, although other mechanisms also exist. Overall, the experiments were carried out carefully and the data were presented very well. Good phenotypic characterization. A major weakness is a lack of mechanistic explanation. Also, these effects on adipogenesis and LD dynamics are fairly common, i.e. manipulating hundreds if not thousands of genes would lead to similar mild effects on adipose tissue as reported here. Also some controls would help.

Specific concerns are listed below.

Comment 2-1. In all overexpression experiments, a nice control would be a known lost-of-function mutant. Is there such a stomatin mutant that abolishes its interaction with cholesterol? What about mutating the proline that is important to the hairpin? Such a mutant should be used in cell experiments.

Response:

Thank you very much for the suggestions. We certainly would like to use “lost-of-function mutants” as controls for the overexpression experiments. To this end, deletion mutant genes have been constructed and introduced to the cells; the effects of over-expressions of these mutant stomatin genes in adipocyte like cells are added as Supplemental Figure 3. We have tried making point mutation of prolin; however, such mutant proteins could not be properly expressed and often seen as aggregates inside the cells.

New texts are added to the Results section to described these new experiments (red texts, third paragraph, lines 215-224, p8):

“Stomatin contains several functional domains. In addition to the wild type, C-terminal truncated mutant (Δ C-hSTOM, 1-263aa) and N-terminal truncated mutant (Δ N-hSTOM, 54-288aa) were constructed (Supplemental Fig. 3A). Over-expression of all these constructs resulted in increase of large LDs (Supplemental Fig. 3B) in adipocyte-like cells; however, the subcellular distribution (data not shown) and fatty acid uptake function of C-terminal truncated mutant was comparable to the wild type (Supplemental Fig. 3C). In contrast, the Δ N-hSTOM displayed puncta-like signals inside the cells and slightly reduced the degree of FA uptake compared to the wild type (Supplemental Fig. 3C). All mutant proteins, like the wild type proteins, could bind to CD36 (Supplemental Fig. 3D).”

Comment 2-2. Figure 1C, immunoEM or Apex-EM is required to better understand stomatin and LDs.

Response:

The immunoEM or Apex-EM experiments, if successfully done, would certainly be very informative. However, the antibodies we have do not allow us to perform these studies. Instead, we had applied STED microscopy that possesses “super-resolution” capability to carefully examined the intracellular localizations of stomatin, focusing on their presence on the LD surfaces. The new experiments’ results are shown in Fig. 1E and described in Results section (red texts, p5):

“Immunofluorescence staining revealed subcellular distributions of stomatin. In addition to puncta staining in the cytosol, stomatin proteins were noted on the plasma membranes (*arrows*), as well as surfaces of LDs (*inset*, Fig. 1C), where they partially colocalized with perilipin proteins when examined under STED microscopy (Fig. 1D). When LDs were isolated from adipocyte-like cells, stomatin was present in the LD fraction, together with the known LD-associated protein perilipin, (Fig. 1E).”

Comment 2-3. Figure 5A. A typical differentiation assay should include qRTPCR of key adipogenic genes such as *pref-1*, *ap2*, *adipsin* etc.

Response:

In response to this comment, new experiments have been done; the results are shown in Supplemental Fig. 6E and described as in Results section (red texts, p12):

“To further validate the microarray results, we performed qPCR experiments on individual genes, focusing on adipogenesis-related genes *Pparg*, *Cebpa*, *Dlk-1*, *Fabp4*, and *Cfd* genes (Fig. 6F); all of them were down-regulated, except for *Dlk-1*.”

Comment 2-4. Mechanism of action. This is a major weakness. Given stomatin is a raft protein, the authors should examine caveolae in adipocytes in KD cells. Both caveolin-1 and cavin 1 are linked to lipodystrophy, and caveolin-1 is a LD protein. Also, CD36 level and localization need to be examined. CD36 localizes to rafts and interact with LDs (see PMID: 32958780). It is likely that stomatin regulate adipogenesis and LD dynamics through caveolae or CD36.

Response:

Thank you very much for the comments. In response to your critiques, new experiments had been performed:

For caveolae and caveolin-1, new results are added in Fig. 3 and Supplemental Fig. 2 and 4. The results are summarized below:

1. We have identified caveolin-1 (CAV-1) as a stomatin-associated protein by immunoprecipitation (Supplemental Fig. 2). CAV-1 is also a LD-associated protein.
2. Decreased expressions of stomatin significantly reduced the content of CAV-1 from the plasma membrane of the cells (Supplemental Fig. 4E).
3. Adding fatty acids to the extracellular medium of adipocyte-like cells did not affect the surface portion CAV-1 (Fig. 3D), nor was the interaction between stomatin and CAV-1 on the cell surface being interfered (Fig 3F).

For CD36, we found that:

1. CD36 is a stomatin-associated protein by immunoprecipitation and Western blotting assays (Fig. 3A and Supplemental Fig. 2). CD36 also localizes to rafts and interacts with LDs.
2. Decreased expressions of stomatin did not affect the content CD36 of the cells (data not shown).
3. Adding fatty acids to the extracellular medium of adipocyte-like cells did not affect the surface portion CD36 (Fig. 3D); however, the interaction between stomatin and CD36 on the cell surface were increased (Fig 3E)

The working hypothesis of stomatin and its potential interactions with CAV-1 and CD36 is depicted in Fig. 9B. See **red texts** on p16 in the Discussion section (*lines 450-466*).

“Although some fatty acids can cross plasma membrane by passive diffusion⁴⁹, most fatty acid uptake is mediated by membrane-associated transporters; many of them reside and function in the lipid rafts, including CD36 and a variety of fatty-acid-binding proteins (FABPs). CD36, also known as fatty acid translocase (FAT), is an integral membrane protein found on the surface of many cell types in vertebrate animals. Long-chain fatty acid (LCFAs) can bind to CD36. The resulting FAT/CD36 may partition into lipid rafts (1, Fig 9B, see also^{50, 51}). Exposure of adipocyte to LCFAs is noted to also relocate stomatin proteins from LDs to the plasma membrane, especially to the lipid rafts (2). In lipid rafts, stomatin can function as an anchor or organizer to initiate, or maintain, the formation of molecular complexes that internalize lipid ingredients from the extracellular environment (3). Stomatin may also modulate the function of effector molecules residing within the lipid rafts^{52, 53} by capturing or trapping the lateral diffusion of proteins and promote their interactions. Other fatty acid binding proteins (FABPs) may also participate in formation of this translocator complex and accelerate the internalization of LCFAs, resulting in an increased production of intracellular triglycerides (TG), that were then transported to, and stored in LDs (4). Stomatin may also involve in the latter process.”

Based on the current results, we cannot rule out the possibility that stomatin regulates adipogenesis and LD dynamics through caveolae or CD36. This consideration is reflected in Discussion section (**red texts**, first paragraph, *lines*

490-497, p17)

“Decreased stomatin expressions also relocated CAV-1 from cell membranes to intracellular compartments (Supplemental Fig. 4B). CAV-1 is the main protein component of caveolae which are flask-shaped invaginations in the plasma membranes, and has been implicated in regulating cellular signal transduction, cholesterol homeostasis⁶¹, and facilitating FA uptake by CD36-mediated caveolar endocytosis³⁶. Whether stomatin’s activities reported in this report related to the caveolae-mediated endocytosis is currently unknown.”

Comment 2-5. What happens to cholesterol in these O/E and KD cells? Cholesterol in adipocytes can impact adipogenesis and LD dynamics (see PMID: 31727739). Given its role in raft, stomatin may regulate adipogenesis and LD dynamics through cholesterol.

Response:

In response to the comment, new experiments were performed and the results are shown in Supplemental Fig. 4, and described in Results section (*red texts, lines 224-231 p8-9*).

“The content and distribution of free cholesterol inside the cell measured by filipin staining (Supplemental Fig. 4A) and internalization of extracellular cholesterol into the cell quantified by uptake of fluorescently-labeled CholEsteryl (CholEsteryl Bodipy 542/563 C11) added to the culture medium (Supplemental Fig. 4B), were not affected by over-expression of wild type or mutant stomatin. Knockdown of stomatin, on the other hand, decreased the free cholesterol content (Supplemental Fig. 4C) and down-regulated cholesterol uptake from outside of the cell (Supplemental Fig. 4D).”

Are the stomatin’s regulations on adipogenesis and LD dynamics related to its effects on cholesterol metabolisms? This notion is discussed in the Discussion section (*red texts, first paragraph, p17*)

“Are above-mentioned stomatin’s functions related to the protein’s modulatory effects on cholesterol contents of the cell? As shown in Supplemental Fig. 2, stomatin-associated proteins include those involved in cholesterol transport, biosynthetic process and homeostasis; so, stomatin appears to be involved in

cholesterol metabolisms. The filipin staining showed that free cholesterols were accumulated on LD surfaces and plasma membranes of adipocytes⁶⁰. Increased expressions of stomatin or its truncated mutants did not affect the amounts and distributions of free cholesterol (Supplemental Fig. 4A), neither was the uptake of cholesterol influenced by excessive wild type or mutant stomatin proteins. On the other hand, stomatin knockdown significantly decreased the cholesterol content in adipocyte-like cells. The regulatory mechanisms for such inhibition are unclear. Decreased stomatin expressions also relocated CAV-1 from cell membranes to intracellular compartments (Supplemental Fig. 4B). CAV-1 is the main protein component of caveolae which are flask-shaped invaginations in the plasma membranes, and has been implicated in regulating cellular signal transduction, cholesterol homeostasis⁶¹, and facilitating FA uptake by CD36-mediated caveolar endocytosis³⁶. Whether stomatin's activities reported in this report related to the caveolae-mediated endocytosis is currently unknown.”

Minor points:

1. The first sentence of the abstract is grammatically incorrect. “is” not “are”
2. The authors need to be careful with the term lipogenesis, which strictly speaking, is the conversion of glucose to fatty acids. Should use TAG synthesis instead.
3. Line 138, membranes of LDs? What does it mean? LDs do not have a membrane as they contain only a monolayer.
4. Ref. 30 is wrong. No one can repeat that data.

Response:

Thank you very much for the corrections.

1. The incorrect grammar in the Abstract has been amended.
2. The use of the term “lipogenesis” is avoided in the revised manuscript.
3. “Membranes of LDs” are changed to “surfaces of LDs” to avoid confusion.
Thank you.
4. The original Ref. 30 has been removed in the revised manuscript.

Reviewer #3 (Remarks to the Author):

This is a highly interesting study on the function of stomatin, an enigmatic, lipid raft-associated, monotopic, integral membrane protein with peculiar structure that is highly conserved during evolution from archaea to humans. The authors found that stomatin is an important player in adipocyte differentiation and lipogenesis; it is involved in fatty acid uptake and growth of lipid droplets (LDs). Transgenic stomatin overexpressing mice, in contrast to control mice, when fed with high fat diet, were obese and insulin-resistant; they showed hepatomegaly, hepatic dysfunction and steatosis. From these data one can clearly infer that stomatin is a major component of adipogenesis.

•What are the major claims of the paper?

(1) The “lipid raft” protein stomatin participates in adipogenesis and lipogenesis by preferentially recruiting effectors to lipid droplets (LDs) to induce fatty acid uptake and LD fusion. Adipocytes with increased stomatin expression exhibit higher levels of fatty acid uptake and LD growth or enlargement.

(2) Transgenic mice expressing human stomatin that were fed with a high-fat diet showed increased stomatin expression that facilitated progression of obesity, caused insulin resistance and hepatic impairments.

(3) Inhibition of stomatin by gene knockdown or pharmacological treatment blocked not only LD growth but also adipogenic differentiation by downregulation of the PPAR γ pathway.

(4) Effects of stomatin on PPAR γ involved ERK signaling; however, an alternate pathway also exists.

(5) Amongst various anti-obesity measures, stomatin serves as another potential therapeutic target.

•Are the claims novel?

The claims are absolutely novel. Stomatin has been identified on LDs before but the functional role of stomatin in adipocytes or on LDs has not been shown previously. Nothing was known about the involvement of stomatin in the PPAR γ pathway. Nothing related has been reported about stomatin knockout mice.

•Will the paper be of interest to others in the field?

Certainly, this paper will be of great interest to researchers in many fields, such as biochemists, pharmacologists, nutrition scientists and general physicians. The global obesity epidemic is one of the major concerns of society and therefore I believe that the paper will be highly interesting also for a wider public and probably also for pharmaceutical companies.

•Will the paper influence thinking in the field?

The paper will have a great impact on the thinking in the field of adipogenesis and the mechanism of LD generation. Stomatin has not been a player in this context before. The paper will also stimulate research in the caveolin field, because caveolins, which are structurally similar, have been identified on LDs but their function on LDs is still unknown.

•Are the claims convincing? If not, what further evidence is needed?

In general, the claims appear convincing. The data appear sound. In detail, the data showing LD-vesicle fusion could be more convincing, particularly the notion that small LDs are fusing with large LDs appears more incidental. It is not known, whether these small vesicles are really LDs.

Response:

Thank you for raising this concern. In response to the critique, new experiments demonstrating functional outcomes of LD-LD fusion (i.e., replenishing of LD content through fusion) were performed (see Fig. 2E). The results are interpreted and discussion in Discussion section (red texts, p15)

“Direct observations of LD fusion are hard to achieve due to either phototoxicity that hinder the fusion biology, or the fact that LDs undergoing fusion are submicron in size and are often indiscernible under light microscopy. However, the outcome of fusion, such as replenishing of LD contents, can be readily measured by FRAP experiments (Fig. 2E).”

• Are there other experiments that would strengthen the paper further? How much would they improve it, and how difficult are they likely to be?

The existence of tiny LDs could be easily verified by immunofluorescence co-

staining with anti-stomatin and anti-perilipin (or another LD marker) and showing co-localization.

Response:

Thank you very much for the suggestion. ImmunoEM or Apex-EM experiments, if successfully done, would certainly be very informative. However, the antibodies we have do not allow us to perform these studies. Instead, we had applied STED microscopy that possesses “super-resolution” capability to carefully examined the intracellular localizations of stomatin, focusing on their presence on the LD surfaces and comparing their localizations with perilipin. The new experiments’ results are shown in Fig. 1E and described in Results section (**red texts**, p5):

“Immunofluorescence staining revealed subcellular distributions of stomatin. In addition to puncta staining in the cytosol, stomatin proteins were noted on the plasma membranes (*arrows*), as well as surfaces of LDs (*inset*, Fig. 1C), where they partially colocalized with perilipin proteins when examined under STED microscopy (Fig. 1D). When LDs were isolated from adipocyte-like cells, stomatin was present in the LD fraction, together with the known LD-associated protein perilipin (Fig. 1E).”

•Are the claims appropriately discussed in the context of previous literature?

In my view, the claims are appropriately discussed in the context of literature. The claim that stomatin serves as a potential therapeutic target, appears somewhat premature.

Response:

We agree with the reviewer’s opinion and have removed “stomatin serves as a potential therapeutic target” from the original manuscript. Thank you for the suggestion.

•Is the manuscript clearly written? If not, how could it be made more accessible?

The manuscript is clear, however, it should be undergone further careful proofreading regarding typos and grammar. On p.9, line 245, the headline says “Somatin” instead of “Stomatin”.

Response:

We thank the reviewer's careful reading. The revised manuscript has been gone through careful proofreading regarding typos and grammar.

•Could the manuscript be shortened to aid communication of the most important findings?

The manuscript is packed with data derived from studies using various methods. I do not see a possibility to shorten any of these studies.

Response:

Thank you.

•Have the authors done themselves justice without overselling their claims?

In my view, the claims made are justified.

Response:

Thank you.

•Have they been fair in their treatment of previous literature?

As far as my field is concerned, they had been fair in citing the relevant literature.

Response:

Thank you.

•Have they provided sufficient methodological detail that the experiments could be reproduced?

The methods are described in detail and should be reproducible.

Response:

Thank you.

•Is the statistical analysis of the data sound?

The statistical analyses appear sound.

Response:

Thank you.

•Should the authors be asked to provide further data or methodological

information to help others replicate their work?

The methods are more or less standard methods. I do not see the necessity for more detailed information.

Response:

Thank you.

•Are there any special ethical concerns arising from the use of animals or human subjects?

Certainly, many mice have been sacrificed during this study, however, at a certain point there is no alternative to animal experiments, particularly in biomedical research.

Response:

Thank you.

Reviewers' Comments:

Reviewer #1:

Remarks to the Author:

The reviewer's concerns have been fully addressed, and I have no further comments.

Reviewer #2:

Remarks to the Author:

The authors have made great effort in addressing my concerns. New experiments have been added and the results made the study more complete and convincing.

A minor concern is on the language. The opening sentence of the abstract reads a bit weird. Some help from native speakers would benefit this paper.